# ON THE ROLE OF POPULATION HETEROGENEITY IN EMERGENT COMMUNICATION

**Mathieu Rita**
INRIA, Paris
mathieu.rita@inria.fr

**Florian Strub**
DeepMind
fstrub@deepmind.com

**Jean-Bastien Grill**
DeepMind
jbgrill@deepmind.com

**Olivier Pietquin**
Google Research, Brain Team
pietquin@google.com

**Emmanuel Dupoux**
EHESS,ENS-PSL,CNRS,INRIA
Meta AI Research
emmanuel.dupoux@gmail.com

## ABSTRACT

Populations have often been perceived as a structuring component for language to emerge and evolve: the larger the population, the more structured the language. While this observation is widespread in the sociolinguistic literature, it has not been consistently reproduced in computer simulations with neural agents. In this paper, we thus aim to clarify this apparent contradiction. We explore emergent language properties by varying agent population size in the speaker-listener Lewis Game. After reproducing the experimental difference, we challenge the simulation assumption that the agent community is homogeneous. We then investigate how speaker-listener asymmetry alters language structure through the analysis a potential diversity factor: learning speed. From then, we leverage this observation to control population heterogeneity without introducing confounding factors. We finally show that introducing such training speed heterogeneities naturally sort out the initial contradiction: larger simulated communities start developing more stable and structured languages.

## 1 INTRODUCTION

Language emergence has been explored in linguistics and artificial intelligence for two main reasons (Lazaridou & Baroni, 2020). On the one hand, artificially reproducing language emergence may help to understand the evolution of human languages (Steels, 1997; Briscoe, 2002; Wagner et al., 2003). On the other hand, language is known to be structured, and compositional (Bickerton, 2007), and imitating such properties would enhance machine learning representations. As a result, there exists a constant back and forth between cognitive sciences, linguistics, and artificial intelligence to retrieve the core ingredients of language emergence (Kirby et al., 2008). In this paper, we explore how the size of a population may impact the structure of emerging languages by using neural reinforcement learning methods.

Especially, we explore the following socio-linguistic hypothesis: larger communities create more systematic languages (Raviv et al., 2019a; 2020). This hypothesis has been supported by a number of ethnographic (Gary Lupyan, 2010) and socio-linguistics (Raviv et al., 2020) observations as well as behavioral studies mimicking language emergence in a controlled setup (Raviv et al., 2019a). Few neural language emergence papers have explored how community size impacts language structure so far, but the available evidence is mitigated at best. Tieleman et al. (2019) observed a small but consistent regularization effect when pairing auto-encoders within a population. Similarly, Cogswell et al. (2019) observed slight improvements in language compositionality with a large population but only reported them in few experimental settings. Finally, Graesser et al. (2019) studied the impact of contact-agents for different population sizes, but they did not observe a correlation between the population size and the convergence speed, success rate, or mutual agent intelligibility.

The following question arises: why does community size not improve language properties in recent emergent communication literature, although it is a key structuring factor in broader linguistics lit-

erature? We argue that recent emergent communication models are limited as they ignore individual learning capacities by working only with homogeneous populations. Consequently, they miss coupling effects emerging from agents' asymmetries. As a result, we hypothesize that community size effects could occur as soon as local heterogeneities are introduced into populations.

In this work, we explore the effects of population size with neural agents in the well-known Lewis referential game (Lewis, 1969). In this game, a speaker describes a hidden object to a listener, which must then reconstruct object properties. Both agents thus need to co-develop a communication protocol to solve the task. The population-based variant of this game randomly pairs one speaker and one listener from their respective communities. The goal is to observe whether increasing the number of agents enhances the communication protocol qualities, e.g. success rate, compositionality, generalization etc. (Kottur et al., 2017; Chaabouni et al., 2020; Lazaridou et al., 2018).

Firstly, we reproduce Lewis reconstruction setting and confirm the experimental difference: when increasing the number of agents, we do not observe improvements over various emergent language metrics. We thus question the current paradigm to model population in the language emergence literature. In particular, all agents are trained uniformly, i.e., their learning speed, capacity, sampling are identical (Tieleman et al., 2019; Cogswell et al., 2019; Fitzgerald, 2019). Secondly, we evaluate the impact of a potential source of model heterogeneity: agents learning speed. We observe that the absolute value of speaker-listener speed is not important, yet their relative value is crucial. We hence shed light on the strong correlation between language structures and agents relative training facilities. Thirdly, we push this reasoning further by distributing learning speeds across the population thus creating heterogeneous populations. We there observe an improvement of language scores combined with a variance reduction when increasing population sizes. In other words, larger communities of neural agents start developing more stable and structured languages when being heterogeneously designed. This observation brings a first stone toward solving the empirical and computational contradiction.

Our experiments partially removed the apparent contradiction between the socio-linguistic observations and the recent emergent communication literature. They illustrate how crucial population training dynamics are in shaping emergent languages and how population heterogeneity may have been underestimated in the recent emergent communication literature. All in all, our contributions are three-fold: (i) we empirically show that the community size is not a structuring factor in language emergence by or in itself in the classic homogeneous Lewis setting; (ii) we give evidences that speaker-listener relative dynamics strongly affects language properties; (iii) we provide the first computational cues to remove the apparent difference between the sociolinguistic literature and recent neural emergent communication works.

## 2 RELATED WORK

**Population size in sociolinguistics.** Population size is a core parameter defining the social environment an agent interacts with. Its impact on language structures has largely been studied on humans (Nettle, 2012; Bromham et al., 2015; Reali et al., 2018; Raviv et al., 2019a) and animals (Blumstein & Armitage, 1997; McComb & Semple, 2005; Wilkinson, 2013). By analyzing 2000 languages (Dryer & Haspelmath, 2013), a clear correlation was drawn between population size and diverse language features, e.g. larger communities tend to develop simpler grammars (Gary Lupyan, 2010; Meir et al., 2012; Reali et al., 2018). As part of their research on the influence of network structures on language emergence (Raviv et al., 2019a;b; 2020), Raviv et al. (2019a) went one step further by arguing that the community size is predictive of language structure and diversity. To do so, they split 150 people into different groups of given community size to isolate confounding factors. While people played a speaker-listener Lewis game, they observe that the greater the community size, the simpler and more consistent the generated language. Here, we intend to test this assumption in the context of neural language emergence. All things considered, we adopt a setting close to Raviv et al. (2019a)'s when computationally modeling human population.

**Populations in experimental emergent communication.** Experimental language emergence has mainly been studied with two methods: behavioral studies (Kegl, 1994; Sandler et al., 2005) and simulations (neural and non neural) (Wagner et al., 2003; Lazaridou & Baroni, 2020). From behavioral studies and non neural simulations, two main approaches have emerged in the past twenty years: experimental semiotics (Galantucci & Garrod, 2011; Garrod et al., 2007) and iterated learn-

ing (Kirby & Hurford, 2002; Kirby et al., 2008; Beckner et al., 2017). According to experimental semiotics studies, languages are mainly subject to an expressivity pressure; they argue that messages should be highly informative to allow communication within a group (Fay & Ellison, 2013). According to iterated learning paradigm, structures emerge from a compressibility pressure; they argue that memory limitations compel messages to become simpler to be easily learned (Tamariz & Kirby, 2015), which is also referred to as transmission bottleneck (Smith et al., 2003). Kirby et al. (2015) then combined those two approaches and show that languages emerge as a trade-off between expressivity and compressivity during cultural evolution. Recently, similar ideas have been modeled in emergent communication frameworks involving neural agents (Ren et al., 2020; Lu et al., 2020). However, seminal non-neural simulations all used diverse optimization methods and models across studies (Wagner et al., 2003); it is hence hard to generalize a global trend of language emergence due to the experimental specific, and sometimes contradictory conclusions. Modern neural agents manage to simplify and standardize agents' modeling, paving the way for holistic models of emergent communication (Lazaridou & Baroni, 2020). Our paper is related to this last set of neural works.

Recent works in emergent communication have been debating the prerequisites to the emergence of language universals such as compositionality (Li & Bowling, 2019; Ren et al., 2020; Lazaridou et al., 2018; Mordatch & Abbeel, 2018; Resnick et al., 2020; Kottur et al., 2017; Choi et al., 2018; Łukasz Kuciński et al., 2020), generalisation (Baroni, 2020; Chaabouni et al., 2020; Hupkes et al., 2020; Denamganaï & Walker, 2020), efficiency (Chaabouni et al., 2019; Rita et al., 2020) or stability (Kharitonov et al., 2020; Lu et al., 2020). Among them, a few works explored how different population structures may impact properties of emergent languages. Inspired by iterated learning methods, Ren et al. (2020); Li & Bowling (2019); Lu et al. (2020) look at language evolution across multiple generations of agent pairs, i.e. population is spread over time. However, we here consider a population where multiple agent-pairs coexist simultaneously within a single generation. There, Graesser et al. (2019) show that community of agents start coordinating their language when at least three agents are present in the community. They later assume that increasing the community size may impact the emergent shared language, but did not observe it in their initial experiments. A similar hypothesis was also made by Bouchacourt & Baroni (2019) while analyzing the influence of symmetric agents in the emergence of a common language. With different research objectives, Cogswell et al. (2019) exhibit a slight compositionality enhancement when increasing population size without the need of language transmission. Analogously, Tieleman et al. (2019) explicitly study community size and display a small but consistent gain of abstraction and structure within speakers' latent representations by increasing population size. Eventually, Fitzgerald (2019) suggest that populations improve generalization compared to single speaker-listener pairs but underlines that there is not a clear correlation between community size and learning properties. We here analyze how population size affects those discussed properties. We align with emergent communication literature and show that naively increasing community size does not consistently improve language properties. We then challenge the homogeneity assumption made in most population designs.

## 3 METHOD

We here describe the different components of language emergence in a population-based Lewis Game, namely, game rules, notations, training dynamics, and evaluation metrics. Finally, we define how we alter population dynamics by asymmetrizing agents and injecting heterogeneities.

### 3.1 RECONSTRUCTION GAME.

**Game Rules:** We study emergent communication in the context of the Lewis reconstruction games (Lewis, 1969). There, a speaker observes all the attributes of an object. The speaker then outputs a descriptive message, which a second agent, the listener, receives. The listener must accurately reconstruct each value of each attribute to solve the task. Both agents are finally rewarded in light of the reconstruction accuracy. Note that another variant of this game requires the listener to retrieve the correct object within a list of distractors, but both settings are inherently similar.

**Game Formalism:** The observed object $v \in \mathcal{V}^K$ is characterized by $|K|$ attributes where each attribute may take $|\mathcal{V}|$ values. We encode the observed object by a concatenation of one-hot representations of the attributes $v_k \in \mathcal{V}$ of the object $v$ for each attribute $k \in K$. For each new run, the set of objects is split into a training set $\mathcal{X}$ and test set. The intermediate message $m \in \mathcal{W}^T$ is a sequence of $T$ tokens, $m = (m_t)_{t=0}^{T-1}$ where each token is taken from a vocabulary $\mathcal{W}$ of dimension

$|\mathcal{W}|$, finishing by a hard-coded end-of-sentence token $EoS$. The speaker and listener are two neural agents respectively parametrized by $\theta$ and $\phi$. The speaker follows a recurrent policy $\pi_\theta$: given an input object $v$, it samples for all $t$ a token $m_t$ with probability $\pi_\theta(m_t|m_{<t}, v)$. We denote $\pi_\theta(m|v)$ the probability distribution of the entire message given an input object $v$. The listener outputs for each $k$ attributes a probability distributions over the values $\mathcal{V}$: $\pi_\phi^k(v_k|m)$. At training time, the speaker message is generated by sampling the policy $m \sim \pi_\theta(\cdot|v)$. At test time we use the greedy message $\hat{m}_t = \arg\max_{\bar{m}} \pi_\theta(\bar{m}|\hat{m}_{<t}, v)$.

**Game Objective:** As in (Chaabouni et al., 2020), we define the listener training goal to be the average of the multi-classification log-likelihood loss per attribute:

$$\mathcal{L}_\phi = -\frac{1}{|\mathcal{X}|.|K|} \sum_{v \in \mathcal{X}} \sum_{m \in \mathcal{W}^T} \sum_{k \in |K|} \pi_\theta(m|v) \cdot \log\left(\pi_\phi^k(v_k|m)\right). \tag{1}$$

In our setting, we want the speaker to optimize the same objective. To do so, we define the speaker game reward as the negative loss of the listener,

$$r_t(v, m_{<t}) = \begin{cases} \frac{1}{|K|} \sum_{k \in |K|} \log\left(\pi_\phi^k(v_k|m)\right) & \text{if } t = T, \\ 0 & \text{otherwise.} \end{cases} \tag{2}$$

Following the gradient policy theorem (Sutton et al., 2000), we maximize the speaker reward by minimizing the following objective over $\theta$:

$$\mathcal{L}_\theta = -\frac{1}{|\mathcal{X}|} \sum_{v \in \mathcal{X}} \sum_{t \in T} \log \pi_\theta^k(m_t|x, m_{<t}) \cdot r_t(v, m_{<t}), \tag{3}$$

where $m$ is sampled according to the speaker policy $\pi_\theta$.

## 3.2 POPULATION-BASED RECONSTRUCTION GAME

We first create a population of $N$ speakers and $N$ listeners, thus obtaining a total number of $2N$ agents. Following (Tieleman et al., 2019), at each step, we uniformly sample one speaker and one listener and pair them together. We then proceed as in the classic one pair Lewis game: both agents play the game with a batch of inputs and receive an optimization step minimizing (Equation 3 & 1). This operation is repeated until convergence, i.e., all speaker-listener pairs have stable losses. While standard, we note that this training procedure relies on strong latent assumptions: (i) each speaker (resp. listener) is uniformly sampled, i.e., there is no preponderant agent within the population, (ii) the communication graph is fully connected and uniform, i.e., all speakers may be paired with all listeners with the same probability, (iii) agents cannot be differentiated, i.e., agents have no information about the identity of their partners, (iv) speakers and listeners are all similar, i.e., there is no difference in the agent definitions nor in the optimization process. Overall, those hypotheses create a homogeneous training setting. In practice, the agents only differ by their initialization and optimization updates, e.g., stochastic agent pairing, game generations, and message sampling.

## 3.3 ASSESSING EMERGENT LANGUAGE IN LEWIS GAMES.

We here introduce various metrics to assess emergent languages structure and quality. To do so, we first need to introduce two distances in the input/object space and in the message space. We define the distance between two objects $v$ and $v'$ as the proportion of distinct attributes computed by $D_{obj}(v, v') = \frac{1}{|K|} \sum_k \mathbb{1}\{v_k \neq v'_k\}$. For the distance between two messages $m$ and $m'$ we use the edit-distance (Levenshtein et al., 1966) and note it $D_{mes}(m, m')$.

**Speakers synchronization:** Within populations, we measure how close speakers' languages are by introducing a distance between the two languages. Given two set of speaker weights $\theta_1$ and $\theta_2$ and their respective language $\mathbf{L}_{\theta_1}$ and $\mathbf{L}_{\theta_2}$,

$$D_L(\mathbf{L}_{\theta_1}, \mathbf{L}_{\theta_2}) := \mathbf{E}_{v \in \mathcal{X}, m_1 \sim \pi_{\theta_1}(.|v), m_2 \sim \pi_{\theta_2}(.|v)}[D_{mes}(m_1, m_2)]. \tag{4}$$

It computes the average distance between two speakers' messages over all the dataset. When considering an entire population of $N$ speakers, we can then compute the synchronization score $\mathbf{r}_{sync}$ by averaging the distance between all pairs of speakers: $\mathbf{r}_{sync} = 1 - \frac{2}{N(N-1)} \sum_{i \neq j} D_L(\mathbf{L}_{\theta_i}, \mathbf{L}_{\theta_j})$.

**Entropy:** To measure language coherence, we study the entropy of the speaker language $\mathcal{H}_\theta$:

$$\mathcal{H}_\theta := \mathbf{E}_{v,m\sim\pi_\theta(\cdot|v)}\big[-\log(\pi_\theta(m|v))\big] = \mathbf{E}_{v\in\mathcal{X}}\left[\sum_{t=0}^{L-1} h(\pi_\theta(m_t|m_{<t},v))\right], \qquad (5)$$

where $h : w \to -w\log(w)$. In our setting, we seek to minimize entropy as it reduces language complexity and denotes language stability (Kharitonov et al., 2020). When $\mathcal{H}_\theta$ is minimal, the speaker uses a unique message to refer to an object. When $\mathcal{H}_\theta$ is high, the speaker generates synonyms to refer to the same object. While some entropy is beneficial with noisy communication channels, they do not improve robustness in our case; it makes the listener's task harder. To ease reading, we display the negative entropy (Neg-Entropy), so all metrics increases correspond to language improvement.

**Topographic Similarity:** We use topographic similarity as a quantitative measure of compositionality (Brighton & Kirby, 2006; Lazaridou et al., 2018). Topographic similarity captures how well similarities between objects are transcribed by similarities in the message space by computing the Spearman correlation (Kokoska & Zwillinger, 2000; Virtanen et al., 2020) between pairwise object distances ($D_{obj}(v_1, v_2)$) and the corresponding message distances ($D_{mes}(m_{v_1}, m_{v_2})$).

**Generalization:** Test accuracy quantifies how well agents generalize to unseen objects. It measures the average reconstruction success on the test set where we define a success when the greedy prediction perfectly matches the input object. The greedy prediction $\hat{v}$ is the one which maximizes the likelihood computed by the listener given the greedy message $\hat{m}$: $\hat{v} = \arg\max_v \prod_k \pi_\phi^k(v_k|\hat{m})$ with $\hat{m}_t = \arg\max_{\bar{m}} \pi_\theta(\bar{m}|\hat{m}_{<t},v)$ the greedy message.

**Stability:** We consider that a metric is stable if it has low variation across the different populations. To do so, we simply measure the stability of a language metric by computing the standard deviation across seeds. In this paper, all experiments are run over six seeds.

## 3.4   ALTERING POPULATION DYNAMICS

We present how diversity can be introduced within a population either by asymmetrizing speaker-listener in minimal size population ($N = 2$), or distributing heterogeneity within larger populations.

**Diversity factor**: We aim to vary specific agent features toward creating diversity in the population and target one core component: the training speed. We control agents' training speed by changing the number of gradient updates. Formally, we introduce the probability $p$ of an agent to be optimized after each iteration of the game. Therefore, the lower $p$, the slower agent training. This approach has two positive features: (i) it neither alters the communication graph nor the agent sampling, preserving other homogeneity hypotheses in the population, (ii) it is more stable than modifying the learning rate as it avoids large destructive updates.

**Control parameters of local asymmetry:** We provide a control parameter to characterize diversity at the scale of a minimal size population ($N = 2$). Noticeably, $N = 2$ is the closer setting to a single-pair of agents where we can compute speakers synchronization. To analyze the effect of speaker-listener training speed asymmetry, we introduce the relative training speed $\rho_{speed} := p^S/p^L$ where $p^S$ (resp. $p^L$) is speaker's learning speed (resp. listener's learning speed).

**Distributing Population Heterogeneity:** To create heterogeneous populations, we characterize every single speaker and listener with individual properties. We then sample them once at the beginning of the training. Formally, when altering training speed, we sample $p_i \sim \text{Log-}\mathcal{N}(\eta_p, \sigma_p)$ for every $i$ agent in the population, where Log-$\mathcal{N}$ is a log normal distribution. In Appendix D.1, we also explore another diversity factor, i.e. agent capacity, to complete our analysis on heterogeneity.

## 4   EXPERIMENTAL SETTING

**Game Properties:** In the main paper, we use $|K| = 4$ attributes with $|\mathcal{V}| = 4$ values. We also report scores with $|K| = 2$ attributes and $|V| = 10$ values in the Appendix F.1 to illustrate that our observation still holds in a different setting. Finally, we use a vocabulary size of $|\mathcal{W}| = 20$ and a message length $T = 10$. For each run, we generate objects in $\mathcal{V}^K$ uniformly at random, and we split them into a train and a test sets, which respectively contain 80% and 20% of the objects.

**Neural architectures:** The speaker first encodes the input object $v$ in a one hot encoding and then processes it with a linear layer. It uses it to initialize a single layer LSTM (Hochreiter & Schmidhu-

Figure 1: Emergent languages properties of homogeneous populations of increasing sizes. Based on socio-linguistics results, all curves should trend upward with population size.

ber, 1997) with layernorm (Kiros et al., 2015) and a hidden size of 128. Finally, the LSTM output is fed to a linear layer of dimension $|\mathcal{W}|$ followed by a softmax activation. The listener is composed of a look-up table of dimension 128 followed by a LSTM with layernorm and a hidden size of 128. Then, for each attribute $k \in \{1, ..., K\}$, we apply a linear projection of dimension $|\mathcal{V}|$ followed by a softmax activation to the last LSTM output.

**Optimization:** For both agents, we use a Adam optimizer with a learning rate of $5e10^{-3}$, $\beta_1 = 0.9$ and $\beta_2 = 0.999$ and a training batch size of $1024$ when optimizing their respective loss. For the speaker, we set the entropy coefficient of 0.02. Finally, we re-normalize the reward by using $\bar{r}(x, w_{<t}) = \frac{r(x, w_{<t}) - \eta_t^r}{\sigma_t^r}$ to reduce the gradient variance (Sutton et al., 2000), where $\eta_t^r$ and $\sigma_t^r$ are respectively the average and the standard deviation of the reward within the batch at each time step.

We use the EGG toolkit (Kharitonov et al., 2019) as a starting framework. The code is available at `https://github.com/MathieuRita/Population`. All experiments are run over six seeds. In all figures, bars show one standard deviation.

## 5 RESULTS AND DISCUSSION

We first reproduce the impact of the population size on language properties in the homogeneous Lewis setting. We then study speaker-listener asymmetry by altering networks training speed, and highlight the importance of relative training speed in shaping language. We finally use training speed as a non-confounding heterogeneity factor when varying the community-size, and thus tackle the initial contradiction.

### 5.1 COMMUNITY SIZE IS NOT ALONE A LANGUAGE STRUCTURING FACTOR

**Language properties are not enhanced by community size.** In Figure 1, we observe that increasing population size does not improve language properties: speaker synchronization and compositionality remain almost constant; entropy and generalization are even deteriorating. Spearman correlations are respectively -0.44, -0.15, -0.52, -0.14, i.e. there is no positive correlation between those metrics and population size. In addition, we do not observe any gain of language stability by increasing population size: increasing community size does not reduce the standard deviation of the language metrics across seeds. Finally, we note that speakers synchronization is high despite using large communication channel, as also observed by (Graesser et al., 2019). Overall, we confirm that community size does not improve the language properties of neural agents.

**The potential pitfall of homogeneity.** We hypothesize that the absence of positive correlation between population size and language properties may be explained by the simplicity of population modeling. The homogeneity assumption seems restrictive as it may lack the inherent diversity of real human communities. Although agents are not identically initialized, they have the same training dynamics in expectation. Consequently, we formulate the subsequent hypothesis: Community size may be a structuring factor as soon as heterogeneity is introduced in populations design.

### 5.2 LANGUAGE PROPERTIES ARE CONTROLLED BY SPEAKER-LISTENER ASYMMETRIES

Before generating heterogeneous populations, we explore the impact of learning speed heterogeneities at the scale of a minimal size population ($N = 2$) in order not to add confounding factors and better intuit population dynamics. As described in Section 3.4, we here modify the learning speed for a minimal size population. Note that we compute this ratio through multiple values of $p^S/p^L$, where $p \in \{0.01, 0.02, 0.04, 0.1, 0.2, 0.5, 1.\}$.

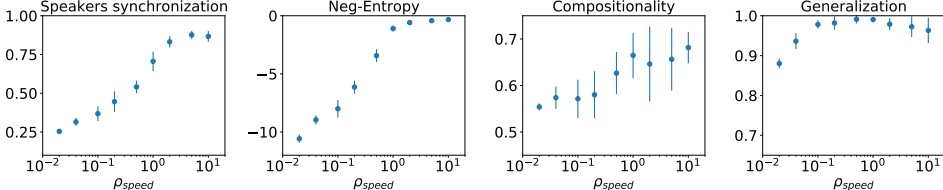

Figure 2: Emergent language properties as a function of speed ratio $\rho_{speed}$ with $N = 2$. Left to right on the x axis, the speaker have higher training speed while the listener have lower training speed.

**Relative differences between speaker and listener significantly affect language properties.** In Figure 2, we display the evolution of language properties with $\rho_{speed}$. We observe that asymmetrizing speaker-listener learning speed significantly influences language properties. When the speakers have a large update probability $p_s$, we note a global improvement of language properties: (i) speakers synchronization rate is very high, (ii) language neg-entropy is reaching almost 0, (iii) compositionality is significantly improved and (iv) test accuracy is close its optimal value. On the contrary, when listeners have a large update probability compared to the speakers, we note the opposite trend. In Appendix C, we also test different game setups for completeness, and obtain similar conclusions.

As a sanity check, we then verify that altering $p$ does not fundamentally change the language properties per itself. We thus sweep over the gradient update probability $p$ while setting a minimal population size of $N = 2$, and compute the Spearman correlation in Table 1. We observe that hyperparameter $p$ has no statistically significant correlations with most of the language scores, i.e. their Spearman coefficients are below $0.4$ with p-values above $0.05$. Therefore, the impact of this correlation is small enough in our experiments to be neglected. In sum, it is not the magnitude of the parameters that is critical to shape language, but the relative ratio between them.

**Understanding learning speed asymmetries.** We here try to provide some pieces of intuitions to understand $\rho_{speed}$ by looking at the extreme cases. In the limit $\rho_{speed} \ll 1$, i.e. $p_s \ll p_l$, the listener is almost optimal wrt. the speaker at each speaker update. It implies that listener's loss is optimally minimized for each message referring to a single input. Then, as soon as speakers develop a language where each message refers to a single input, the game is over as speaker's rewards are immediately maximal. It means that many languages can emerge from this limit case, including almost degenerated languages.

In the limit $\rho_{speed} \gg 1$, i.e. $p_s \gg p_l$, the speaker is almost optimal wrt. the listener at each listener update. The speaker thus targets the messages providing the highest rewards as in a stationary RL task. We can safely assume that the set of messages providing the highest rewards for each input is small. It then explains why the speaker converges to a low entropy language. In addition, when $N = 2$, both speakers solve the same communication task on same quasi-stationary environment and obtain similar rewards, they are then more likely to align on a similar language. This use-case also suggests that when speakers are "fast enough", a common interlocutor is the only requirement for the emergence of a common language. Finally, we push further the assumption by interpreting the gain of compositionality through "ease-of-teaching" (Li & Bowling, 2019). The authors show that compositional codes are easier to teach for listeners. We then hypothesize here that 'faster' speakers develop codes easier to learn for 'slow' listeners and thus higher quality languages.

In Appendix D.1, we show that trends noticed in Figure 2 can be reproduced varying the ratio between speaker's and listener's capacities. It suggests that other parameters influencing the relative speaker-listener co-adaptation affect language properties. We provide first hints on an indirect influence of this capacity ratio on agents' relative training speed. More generally, we suspect that many parameters may be related to training speed, e.g. community-graph (Graesser et al., 2019), newborn agents (Ren et al., 2020; Cogswell et al., 2019), or broadcasting by using different speaker-listener ratios as illustrated in Appendix D.2, but we leave it for future analysis.

Table 1: Spearman Correlation between $h$, $p$ and language scores. p-values $< 0.05$ are underlined.

| Param. | Sweep | Sp. sync | Neg-Entropy | Compo. | Gene. |
|--------|-------|----------|-------------|--------|-------|
| $p$ | {0.01, 0.02, 0.04, 0.1, 0.2, 0.5, 1. } | 0.13 | 0.33 | 0.05 | -5e-3 |

Figure 3: Emergent languages properties of heterogeneous populations of increasing sizes. For heterogeneous populations, we start recovering the socio-linguistics correlation. Orange dashed line shows homogeneous populations and green dashed line the best pair for $N$=2 across all $\rho_{speed}$.

## 5.3 COMMUNITY SIZE STRUCTURES LANGUAGES OF HETEROGENEOUS POPULATIONS

Previous sections have identified that learning speed is a powerful controlling factor and could alter language structure when asymmetrizing speaker-listener pairs. However, these experiments remain limited for (i) we have not yet observed larger population size tends to generate more stable and structured language, (ii) this asymmetry remains artificial and specific to our Lewis game setting. Therefore, we here leverage this training speed factor within a population, characterizing individual agents with different update probabilities. We hence sample agent update probability with $p \sim$ Log-$\mathcal{N}(\eta_p, \sigma_p)$ with $\eta_p = -1$ and $\sigma_p = 1$ if not specified otherwise.

**Larger heterogeneous population leads to higher quality language.** In Figure 3, we observe that emergent language metrics are now positively correlated with population size. When increasing the population size from 2 to 20 speaker synchronization and neg-entropy go from low to almost optimal values, compositionality increases by 22% in average and generalization remains stable. As detailed in Appendix E, speaker synchronization, neg-entropy and compositionality have a significant correlation with Spearman coefficients equal to 0.55, 0.24, and 0.21 (p-values< 0.05). Those are the first hints toward our initial goal: when individuals are heterogeneous, a large population size correlates with higher quality and stable language properties. In Appendix F.2, we report the results with distribution changes. When the distribution is more concentrated, trends are less significant and we note that metrics do not have the same sensitivity to heterogeneities: a minimal dispersion of the distribution is required to observe significant effects. When changing the distribution without changing variance ($\beta$ distribution instead of log-normal distribution), trends are not notably affected.

The overall gain over language metrics can be interpreted in light of Section 5.2. Language score evolution depends on the relative speaker-listener learning speeds. In heterogeneous populations, the probability of sampling extreme learning speed $p$ and thus enforcing speaker-listener asymmetries increases with $N$. One explanation would be that extreme agents, especially slow listeners, behave as kinetic bottlenecks, forcing the speakers to structure their languages. Thus, language emergent properties would be particularly determined by fast speakers and/or slow listeners.

**Larger heterogeneous population leads to more stable language.** Similarly, the population size correlates with the variance of the language scores across seeds. Hence, when increasing the population size from 2 to 20; the standard deviations of speakers synchronization, neg-entropy, compositionality are divided by 10, 15 and 2; yet generalization is less affected. One hypothesis for this reduction is that smaller communities are more likely to be disparate because of the sampling. For instance, if we characterize a population with the empirical average of the training speeds $\hat{\eta}_p$, it has high variance with small $N$. On the other side, when $N$ grows, $\hat{\eta}_p$ gets closer to the average learning speed, $\lim_{N \to \infty} \hat{\eta}_p = \eta_p$, and its variance gets smaller. In other words, larger heterogeneous communities have more stable languages because of Law of large numbers in our models.

**Increasing diversity enforces population size benefits.** In Figure 4, we challenge further the impact of heterogeneity by varying the standard deviation $\sigma_p$ of the learning speed distribution. For each value of $\sigma_p$, we run the game with populations of size $N = 2$ and $N = 10$ and compare the relative evolution of the scores between the two population sizes. We note that the higher $\sigma_p$, the larger the gain of synchronization, neg-entropy, generalization when increasing $N$. The relative gain of neg-entropy is almost equal to 100% when $\sigma_p > 1$ while speaker synchronization gain is close to 200% for large values of $\sigma_p$. For generalization, the 10% relative gain is noteworthy as it corresponds to reaching close to 100% test accuracy. The more discrepancy exists within the population

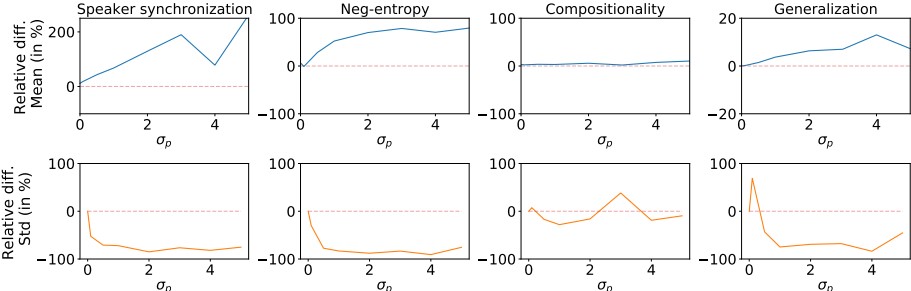

Figure 4: Relative variations in average language scores (top, blue line) or standard deviations (bottom, orange line) between populations of size $N = 10$ and $N = 2$ as a function of increasing heterogeneity $\sigma_p = \{0, 0.1, 0.5, 1, 2, 3, 4, 5\}$. It is defined by $100 \times (\bar{y}_{N=10} - \bar{y}_{N=2})/\bar{y}_{N=2}$ where $\bar{y}_N$ is the average of the mean (or std) across seeds for a population of size $N$. The higher diversity the better the benefits: language is of higher quality (blue), and the language is more stable (orange).

(large $\sigma_p$), the more beneficial is to have large populations. Furthermore, speaker synchronization, neg-entropy and generalization standard deviations reduce up to almost 80% when having $\sigma_p > 2$. In sum, this result corroborates that larger populations lead to more stable language.

**On the overall performances of heterogeneous populations.** On Figure 3, we see that language properties of homogeneous populations with $\rho_{speed} = 1$ decrease when population size is increasing. On the contrary, metrics get higher for heterogeneous populations (generalization stays almost constant). Thus, when modeling large populations, there is a gain distributing heterogeneities within populations. However, metrics remain below the best pair observed for $N = 2$ when varying $\rho_{speed}$. It means that enlarging populations does not lead to an absolute better language than the best one we can obtain with a minimal size population. Though, when population size increases, heterogeneous populations metrics get closer to the best pair scores. Therefore, population size tends to synchronize agents on a language whose properties are close to those of the best single pair.

## 6   CONCLUSION

One objective of language emergence simulations is to identify simple (non-biological) rules that characterize language evolution (Steels, 1997; Briscoe, 2002; Wagner et al., 2003; Kirby, 2001). However, those models require a delicate balance between simplicity to ease analysis and complexity to remain realistic. In this paper, we argue that the current population-based models may actually be too simplistic as they assume homogeneous communities. We show that this simplification is a potential root cause of the experimental difference between neural observations and the sociolinguistic literature. Namely, larger populations should lead to more stable and structured languages, which is not observed in recent emergent models. Yet, as soon as we add diversity within populations, this contradiction partially vanishes. We advocate for better integrating population diversity and dynamics in emergent literature if we want computational models to be useful. In this journey, we also observe that the relative training speed of the agents may be an underestimated factor in shaping languages. Worse, other agent properties may be confounded with it, such as network capacity.

This work also opens many questions. First, our observations were performed in the Lewis game so far, and it would be interesting to observe the impact of diversity in more complex settings. Second, while heterogeneity is a structuring factor in population, the core factors are yet to be investigated, e.g., how to correctly model diversity, are there some causal components? Third, we break the homogeneous assumptions in our work, but other procedures may exist that solve the initial experimental difference such as varying the communication network topology (Wagner, 2009), the proportion of contact agents (Wray & Grace, 2007; Clyne, 1992; Graesser et al., 2019) or the proportion of second learners (Ellis, 2008). Fourth, although heterogeneous populations recover a sociolinguistic result, the average scores remain below the best emergent protocol. We leave for future work how heterogeneous populations may be leveraged to structure further the language (e.g. more complex tasks, larger population). Finally, training speed is a natural controlling parameter for computational models, but it is unclear how it may relate to human behavior. Overall, we hope that this paper provides new insights toward modeling populations in language emergence, and it is part of this constant back and forth between cognitive sciences, linguistics, and artificial intelligence.

## ACKNOWLEDGMENTS

Authors would like to thank Corentin Tallec, Rahma Chaabouni, Marco Baroni, Emmanuel Chemla, Paul Smolensky, Abdellah Fourtassi, Olivier Tieleman, Kory Mathewson for helpful discussions and the anonymous reviewers to their relevant comments. M.R. was supported by the MSR-Inria joint lab and granted access to the HPC resources of IDRIS under the allocation 2021-AD011012278 made by GENCI. E.D. was funded in his EHESS role by the European Research Council (ERC-2011-AdG-295810 BOOTPHON), the Agence Nationale pour la Recherche (ANR-17-EURE-0017 Frontcog, ANR-10-IDEX0001-02 PSL*, ANR-19-P3IA-0001 PRAIRIE 3IA Institute) and grants from CIFAR (Learning in Machines and Brains) and Meta AI Research (Research Grant).

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

## A    RANDOM BASELINES

We report the values of speakers synchronization, neg-entropy, compositionality (topographic similarity) and generalization (test accuracy) for randomly-initialized and untrained agents:

Table 2: Average speakers synchronization, neg-entropy, compositionality and generalization for randomly-initialized and untrained agents. We report the standard deviation across 10 seeds

| Experiment | Sp. sync | Neg-Entropy | Compo. | Gene. |
|---|---|---|---|---|
| Random agents | $0.09 \pm 0.03$ | $-28.4 \pm 0.23$ | $0.04 \pm 0.02$ | $0.24 \pm 0.02$ |

## B    NON-CONFOUNDING PARAMETERS

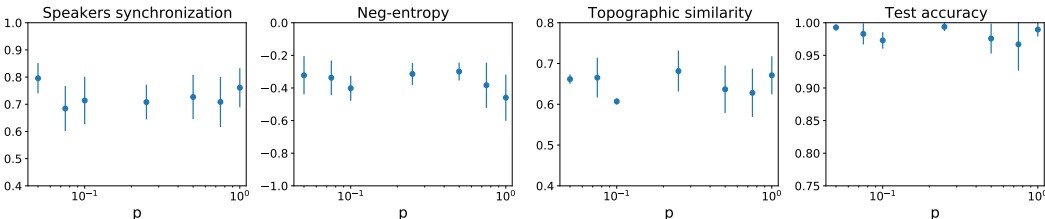

Figure 5: Variations of language metrics with increasing values of gradient update probability $p$.

In Figure 5, we show the variations of language scores with increasing values of speaker probability update $p_S$ and listener probability update $p_L$, while keeping them identical ($p = p_S = p_L$). In Table 1, we show the Spearman correlation between the metrics and $p$. We do not observe significant correlation. In addition, both the relative and absolute variations of the parameters are negligible compared to experiments displayed in Figure 2.

## C    VARYING SETUP PARAMETERS WHEN STUDYING SPEAKER-LISTENER ASYMMETRIES

In this Section, we show how trends displayed in Figure 2 evolve when performing some changes of parameters in the setup. As shown in Figure 6, when modifying agents' hidden sizes (speaker's hidden size is equal to 64 and listener's hidden size is equal to 512 while they are both equal to 128 in Figure 2), the trends are the same. However, we notice a shift of the curves along the x-axis. It means that the ratio of learning speed at which metrics start becoming optimal is controlled by the parameters of the setup. Figure 6 highlights that the trends are sensitive to agents' capacity.

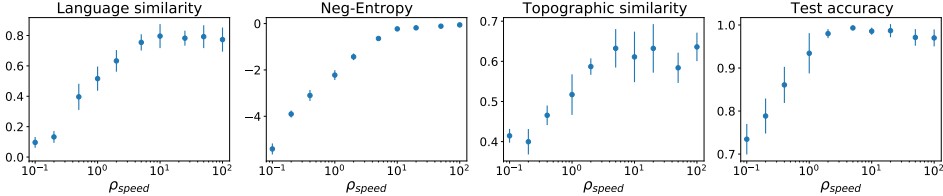

Figure 6: Emergent language properties as a function of speed ratio $\rho_{speed}$ with $N = 2$. From left to right on the x-axis, the speaker have higher training speed while the listener have lower training speed. Compared to Figure 2, we have changed the initial parameters of the problem by modifying agents' hidden sizes. Here: speakers' hidden size is equal to 64 and listeners' hidden size is equal to 512

# D OTHER SPEAKER-LISTENER ASYMMETRIES INFLUENCING EMERGENT LANGUAGE PROPERTIES

In complement of Section 5.2, we show in this Appendix that trends observed in Figure 2 can be reproduced with other speaker-listener asymmetry parameters. In particular, we study the ratio between speaker's capacity and listener's capacity in a minimal size population ($N = 2$) in Appendix D.1 and the ratio between the number of speakers and listeners in Appendix D.2.

## D.1 AGENTS' CAPACITY

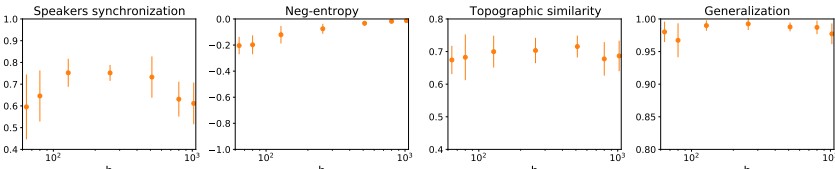

Figure 7: Variations of language metrics with increasing values of agents hidden size $h$.

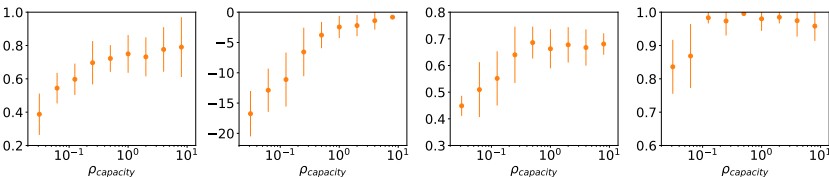

Figure 8: Emergent language properties as a function of capacity ratio $\rho_{capacity}$ with $N = 2$. From left to right on the x-axis, the speaker have higher capacity while the listener have lower capacity.

We here reproduce the trends observed in Figure 2 with another speaker-listener asymmetry. Formally, we introduce $\rho_{capacity} := h^S/h^L$ where $h^S$ (resp. $h^L$) is speaker's hidden size (resp. listener's hidden size). For the following experiments, we compute this ratio with multiple values of $h^S/h^L$ for $h \in \{64, 128, 256, 512, 1024\}$.

**The absolute capacity of agents has no significant impact on language properties.** As done in Section 5.2, we first perform a sanity check and verify that altering $h^S$ and $h^L$ while keeping $h^S = h^L =: h$ does not fundamentally change the language properties per itself. We sweep over $h$ while setting a minimal population size of $N = 2$, and compute the Spearman correlation in Table 3. We observe that $h$ have no statistically significant correlations with speaker synchronization, compositionality and generalization (Spearman correlation inferior to 0.3 or p-values above 0.05). The only noteworthy correlation is between $h$ and the neg-entropy with a Spearman correlation of 0.89 and a p-value$< 0.05$. However, as shown in Figure 7 the variations are two orders of magnitude lower than the neg-entropy variations in Figure 8. Therefore, the impact of this correlation is small enough in our experiments to be neglected.

Table 3: Spearman Correlation between $h$, $p$ and language scores. p-values $< 0.05$ are underlined.

| Param. | Sweep | Sp. sync | Neg-Entropy | Compo. | Gene. |
|---|---|---|---|---|---|
| $h$ | $\{64, 128, 256, 512, 1024\}$ | -0.14 | 0.89 | 0.29 | 0.11 |

**The relative speaker-listener capacity significantly affects language properties** In Figure 8, we see that when varying $\rho_{capacity}$, we get trends similar to those observed in Figure 2. In Table 5, we compute the Spearman correlation between the metrics and $\rho_{capacity}$ and notice that there is a significant correlation between $\rho_{capacity}$ and studied metrics.

**Network capacity may be confounded with training speed.** While the effect of training speed may be partially intuited in Section 5.2, the impact of capacity is more tedious to analyze. As we have already observed that training speed is a crucial parameter, we thus test whether a correlation

between network capacity and learning speed can be established. In Figure 9 (resp. Figure 10), we pair a fixed pretrained listener (resp. speaker) with newly initialized speakers (resp. listeners) of increasing capacities and see how fast those new speakers (resp. listeners) reach convergence. We observe that all trained agents converge to the same accuracy. However, we also note that they have different convergence speeds. Speakers (resp. listeners) of capacity 512, 128, 32 respectively have to see 100, 250, and 900 batches (resp. 50, 100 and 250 batches) to reach train accuracy of 95%. In short: the larger the network, the faster the training speed. We conclude that network capacity may be partially explained by the indirect impact of capacity on learning speed.

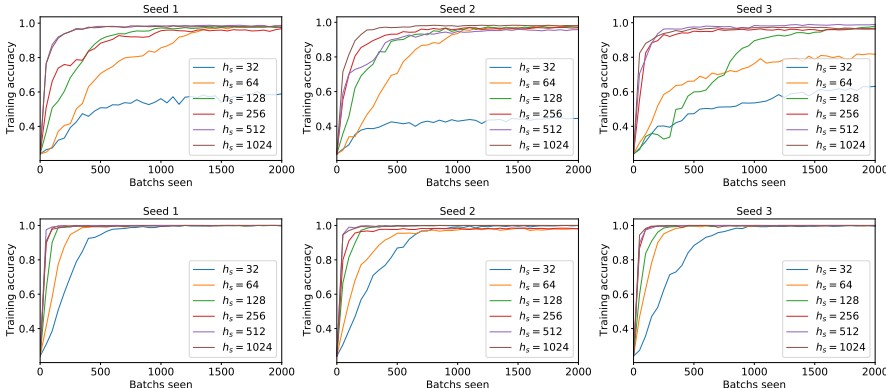

Figure 9: Training accuracy of speaker-listener pairs where the listener is pretrained and fixed. For all curves, a listener has been pretrained with a random speaker, frozen and paired with new initialized speakers of increasing sizes. Each row corresponds to a new pretrained listener ; each column to a new initialization of the speakers. Please note that some limit cases appear with the pure RL task, i.e. fixed listener. Speakers' convergence become unstable across seeds for small networks ($h_S = 32$). Yet, the training is still stable in supervised task, i.e. fixed speaker, in Figure 10.

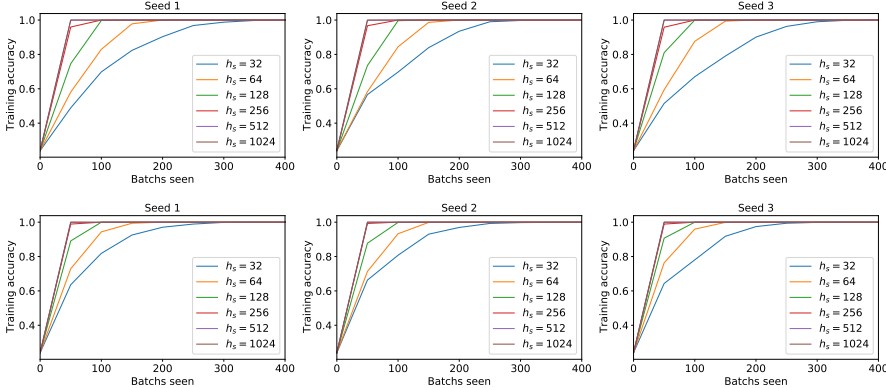

Figure 10: Training accuracy for speaker-listener pairs where the speaker is pretrained and fixed. For all curves, a speaker has been pretrained with a random listener, frozen and paired with new initialized listeners of increasing sizes. Each row corresponds to a new pretrained speaker ; each column to a new initialization of the listener

## D.2    ALTERING THE RATIO BETWEEN THE NUMBER OF SPEAKERS AND LISTENERS

In complement to $\rho_{speed}$ in Section 5.2 and $\rho_{capacity}$ in Appendix D.2, we extend the results to another population factor and intuit how it relates to learning speed. We introduce the ratio between the number of speakers and the number of listeners within a population $\rho_{agents} := \frac{N_{listeners}}{N_{speakers}}$. Note that this ratio cannot be applied to a single speaker-listener pair. Yet, it introduces an asymmetry between the community of speakers and the community of listeners. We show the evolution of

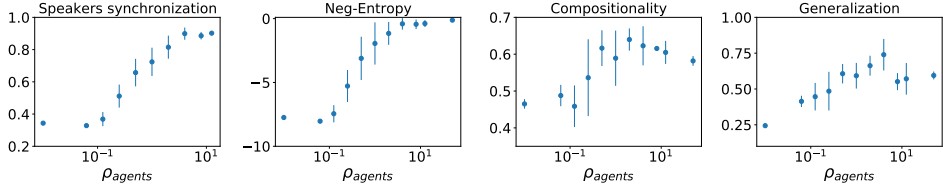

Figure 11: Language metrics for different ratio between $N_{speakers}$ and $N_{listeners}$ where $N_{speakers}$ and $N_{listeners}$ are in the range $\{1, 2, 4, 8, 16, 24, 50, 75, 100\}$. Experiments are run with $|K| = 2$ and $|V| = 10$.

language metrics in Figure 11. There, we observe than unbalancing $N_{speakers}$ and $N_{listeners}$ has a strong impact on language scores. Speaker synchronization reach upper and lower bound depending on the ratio, neg-entropy either is low or close to 0, and compositionality and generalization increase. $\rho_{agents}$ is thus an additional control factor of language properties. Extreme cases coincide with the training speed interpretation of Section 5.2. Indeed, at each iteration of the game, each speaker (resp. listener) has a probability $1/N_{speakers}$ (resp. $1/N_{listeners}$) to be sampled. Thus, speakers have $N_{listeners}/N_{speakers}$ more of less learning steps than listeners. Consequently, when $\rho_{agents} \gg 1$, individual speakers (resp. listeners) have way more optimization steps than individual listeners (resp. speakers) and we fairly hypothesize that speakers (resp. listeners) train faster than listeners (resp. speakers). This ratio may therefore be a confounding factor of a training speed mechanism. Note that, interestingly, $\rho_{agents}$ can also be related to different sampling procedures, or broadcasting phenomena.

## E ADDITIONAL CORRELATIONS

Table 4: Spearman Correlation between community size and language scores for homogeneous and heterogeneous populations ($\eta_p = -1$, $\sigma_p = 1$). These correlations are related respectively to Figure 1 and Figure 3. We underline correlation with $p < 0.05$.

| Experiment | Sp. sync | Neg-Entropy | Compo. | Gene. |
|---|---|---|---|---|
| Homogeneous population | -0.44 | -0.52 | -0.15 | -0.14 |
| Heterogeneous population | 0.55 | 0.24 | 0.21 | -0.04 |

Table 5: Spearman Correlation between controlling factor ratios and language scores in Figure 2. We underline correlation with $p < 0.05$.

| Param. | Sp. sync | Neg-Entropy | Compo. | Gene. |
|---|---|---|---|---|
| $\rho_{speed}$ | 0.96 | 0.97 | 0.63 | 0.34 |
| $\rho_{capacity}$ | 0.49 | 0.77 | 0.33 | 0.09 |

## F COMPLEMENTARY SETTING FOR POPULATION RESULTS

### F.1 DIFFERENT INPUT SPACE

We first reproduce our core experiments in a different Lewis game setting, with $|K| = 2$ attributes containing each $|V| = 10$ values. Results are displayed in Figure 12 and Figure 13 that respectively reproduce the results of Figure 1 and Figure 3. We observe that the main trends observed in the main paper are unchanged when varying parameters of the input space.

### F.2 DIFFERENT HETEROGENEITY DISTRIBUTION

We then vary the distribution of agent update probability $p$ within the population to evaluate how the observed trends are sensitive to the heterogeneity distribution.

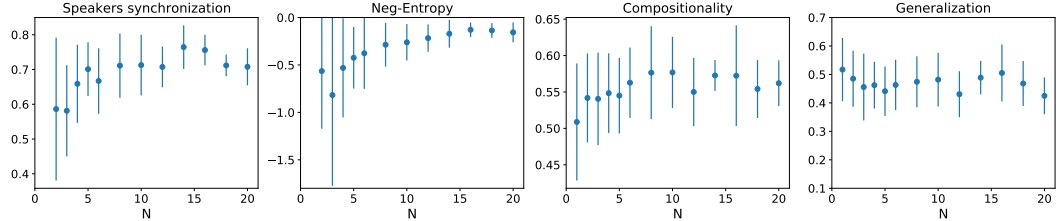

Figure 12: Language scores with homogeneous population with $N = \{1, 2, 4, 6, ..., 20\}$, $|K| = 2$ and $|V| = 10$. Corresponding figure with $|K| = 4$ and $|V| = 4$ in the main paper is Figure 1

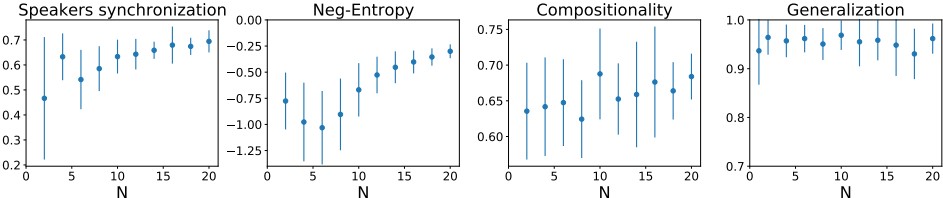

Figure 13: Language scores with heterogeneous population with $N = \{1, 2, 4, 6, ..., 20\}$, $\eta_p = -1$, $\sigma_p = 0.5$, $|K| = 2$ and $|V| = 10$. Corresponding figure with $|K| = 4$ and $|V| = 4$ in the main paper is Figure 3

**Log-normal distribution with lower standard deviation**   We first test how language properties are affected by a variance reduction. In Figure 14 agent update probability $p$ are sampled according to a log-normal distribution Log-$\mathcal{N}(\eta_p, \sigma_p)$ with $\eta_p = -1$ and $\sigma_p = 0.5$. Compared to Figure 3, standard deviation has been divided by 2 meaning a more concentrated heterogeneity distribution. We see that speakers synchronization and neg-entropy increases are slower and that compositionality increase is now very low. Generalization remains almost unchanged. This plot suggests that the distribution of heterogeneities must be minimally wide to observe a positive correlation with population size and that all metrics do not have the same sensitivity to heterogeneities.

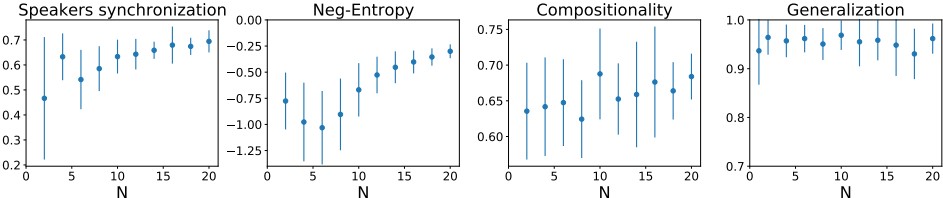

Figure 14: Emergent language properties of heterogeneous populations of increasing sizes. Distribution of agent update probability $p$ is Log-$\mathcal{N}(\eta_p, \sigma_p)$ with $\eta_p = -1$ and $\sigma_p = 0.5$

**Distribution change**   We now test how language properties are affected by a distribution change. In Figure 15, agent update probability $p$ are sampled according to a Beta distribution $\beta(1, 2)$. Variances of $\beta(1, 2)$ and Log-$\mathcal{N}(-1, 0.5)$ (Figure 14) have the same order of magnitude. As we can see, trends are similar for Figure 14 and Figure 15. Trends are thus not notably affected by changing the log-normal distribution by a $\beta$ distribution of similar variance.

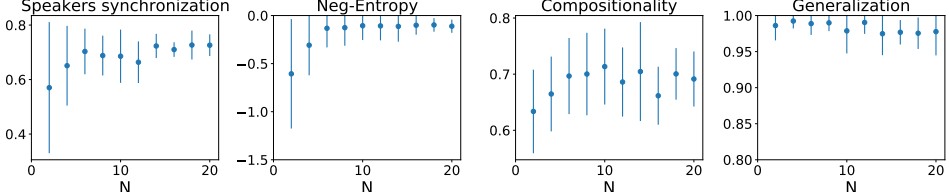

Figure 15: Emergent languages properties of heterogeneous populations of increasing sizes. Agent update probability $p$ follows a $\beta$ distribution $\beta(1, 2)$

