# OpenReview forum: "On the role of population heterogeneity in emergent communication"
_ICLR.cc/2022/Conference — ICLR 2022 Poster_

### Official Review · Reviewer_grQ9 · 2021-10-18

**Correctness:** 3
**Technical Novelty And Significance:** 3
**Empirical Novelty And Significance:** 3
**Recommendation:** 6
**Confidence:** 4

**Main Review:**


# Main assessment
This work is interesting and the topic well worth exploring. The contribution to the area of deep language emergence is very clear and timely. That being said, I do have some concerns. First, I think much more care needs to be put in positioning this paper in the literature.  There is a large body of (non-neural) research on the emergence of compositionality, structure, and generalization that emphasizes that a "transmission bottleneck" (e.g., agents not having the capacity to fully observe/learn the language of their community) is what makes these properties emerge. After all, that's why these properties are useful in the first place: to overcome agents' limitations when learning/imitating/using language. Similar arguments have been put forward for other properties of natural languages, such as ambiguity and vagueness. The authors' main argument follows the same line of reasoning but does not engage with this literature, nor synthesize what we already learned from it. A good starting point is offered by the literature on iterated learning, some of which the authors cite but for other purposes. Another starting point is given by the research that more closely follows Lewis' original setup (e.g., work by and building on "Signals" by Skyrms 2010; or Wagner's "Communication and Structured Correlation"). Second, and more importantly still, while I found the narrative of the paper convincing, I also found the main results rather weak. The bars in Figure 4 strongly overlap. This suggests that there is so much variation between runs that the means are not very trustworthy to draw strong conclusions from. The correlations for heterogeneous populations in Appendix C's Table 2 are analogously weak. I worry that noise may be at play; and an unjustified significance threshold of 0.05 does not dispel this concern.

# Minor comments
There are some issues with the terminology used. The main issue the paper addresses is not a paradox (p. 1) nor is there a contradiction (p. 5).
 The question why population size predicts
structure and complexity in natural language but not in neural language emergence is interesting. However, there is nothing paradoxical or contradictory about it. At most, one may find this puzzling. Similarly "predictive" does not need to mean "causal" (p. 2) and "particularly significant" is a vacuous term (p. 9). This may sound nit-picky but these are all technical term.

I could not quite follow all the details of the equations in Section 3. Part of the reason is that not all symbols are clearly defined.
 For instance, from the text above Equation (1), it's unclear whether T is a set (since its cardinality, |T| is mentioned) or a number (which is what I am led to believe from the formulae that follow, or the text [e.g., p. 5]).
 The equations look reasonable to me but it would be nice if this part of the manuscript was easier to follow.

**Summary Of The Paper:**

# Summary
The manuscript "On the role of population heterogeneity in emergent communication" addresses the question why results on population size in deep language emergence have not, so far, mirrored the effects it is claimed to have on natural language. In a nutshell, in natural communication population size correlates with simpler grammars and less idiosyncratic languages. The effects of population size on the structure of neural emergent communication is less explored and has hitherto not reflected what we know about natural language. The authors argue that one of the reasons for this is that artificial populations are often homogeneous. They show, through simulations, that introducing asymmetries in the training speed of speakers and listeners leads to trends that are more in line with natural language: the size of heterogeneous populations weakly correlates with more aligned languages; higher neg-entropy; better generalization when communicating about novel objects; and more compositional languages (measured as topographic similarity).


**Summary Of The Review:**

This manuscript is on the target. It is interesting and the research is well worth pursuing. However, it has crucial issues that need to be addressed to better appreciate its contributions and scope. In particular, the results do not lend much support to an issue that has already received attention in related (non-neural) research on language emergence.


# Recommendation update

In light of the authors' changes and discussion, I have updated my recommendation. The literature review is much more complete now, and I think weakening some of the statements has made the contribution much clearer. I am still concerned about the rather weak effects and correlations reported, but now the framing is much more in line with the content.

---

> ### Author Response · Authors · 2021-11-18
> **Response to Reviewer grQ9 (1/1)**
>
> *We would like to thank the reviewer for opening the discussion. As far as we understand, there are two main concerns (i) we do not link enough our work on population heterogeneity with the transmission bottleneck hypothesis and pre-neural litterature, (ii) the experimental results do not support our claims and our narrative enough.*
>
> As we think they are the main concerns about the paper and that some points are shared between the reviewers, we provide comments and responses about those concerns in the General Response. Concerning literature, we hope our General Response will enable us to find a scientific agreement.
>
> (I) Concerning the significance of the results of Figure 4, we first clarify the interpretation of the trends for speakers synchronization and neg-entropy to be sure that we agree with Reviewer grQ9. Then, we propose in the General Response to reformulate the contributions and discussion of our results (in addition to the extra experiment suggested by reviewer Rn9H).
>
> (II) Eventually, we provide comments on the minor remarks .
>
> ## I - Clarification of Figure 4
>
> Concerning the significance of the results, we want to clarify what could be expected from the results of Figure 4 and check that we agree on it. Otherwise, it means that we should be clearer in the main text.
>
> We would like to argue that the results of Figure 4 for speaker synchronization and neg-entropy show significant and important effects. Indeed, we observe the following trends: when population size is small, randomness is at play and we indeed see a large variation of the results. Actually, it is an expected effect. We show in Section 5.2 - Figure 3 that speaker synchronization and entropy can be either very high or very low depending on the relative learning speed between agents within a pair. Since we sample variable learning speed ratios when population size is small, this variability is observed. Therefore, having large bars for small pop. size is an expected effect.
>
> On the contrary, when population size is increasing we observe two interesting effects:
>
> - The standard deviations across the seeds become low when pop. size is increasing suggesting that language properties become less variable.
>
> - In addition, the average mean of the scores is getting higher when pop. size is increasing. It means that, as soon as population size is high, the probability to get a community where speaker synchronization is low and neg-entropy is low becomes almost 0. Therefore, pop. size stabilizes speaker synchronization and neg-entropy to high values.
>
> Concerning the statistical analysis, scores are not very impressive (0.35/0.37 for the original Figure 4). We interpret those values by the use of Pearson correlation score that measures a linear correlation. We put it to have an indication of the trends, however, we cannot expect high scores because metrics reach an almost optimal bound quite early when population size is increasing.
>
> __-> Is that what you understood about the results for speaker synchronization and entropy ? If not, we should be clearer in the paper.__
>
> Concerning compositionality and generalization, we acknowledge that Figure 4 may not be strong enough to state that there is an increase of compositionality and generalization with population size. As seen above, the suggestion of Reviewer  Rn9H of increasing heterogeneity yields stronger effects, at least for compositionality __(see the general response for more details)__.
>
> ## II - Minor comments
>
> > There are some issues with the terminology used. The main issue the paper addresses is not a paradox (p. 1) nor is there a contradiction (p. 5). The question why population size predicts structure and complexity in natural language but not in neural language emergence is interesting. However, there is nothing paradoxical or contradictory about it. At most, one may find this puzzling.
>
> “contradiction” and “paradox” have been replaced by “difference”/”experimental difference”/”question” in order to be more factual.
>
> > Similarly "predictive" does not need to mean "causal" (p. 2) and "particularly significant" is a vacuous term (p. 9). This may sound nit-picky but these are all technical term.
>
> Following your comment:
> - we removed “i.e. is a causal factor”
> - we changed the sentence to be more precise
>
> __See the differences in Appendix G5.1 ; G5.2.__
>
> > I could not quite follow all the details of the equations in Section 3. Part of the reason is that not all symbols are clearly defined. For instance, from the text above Equation (1), it's unclear whether T is a set (since its cardinality, |T| is mentioned) or a number (which is what I am led to believe from the formulae that follow, or the text [e.g., p. 5]). The equations look reasonable to me but it would be nice if this part of the manuscript was easier to follow.
>
> |T| is a typo error. We changed it by T which is the number of tokens (or maximal length). For the other notations, |.| effectively is the cardinal of the set.

---

### Official Review · Reviewer_5n9H · 2021-10-30

**Correctness:** 3
**Technical Novelty And Significance:** 4
**Empirical Novelty And Significance:** 4
**Recommendation:** 6
**Confidence:** 2

**Main Review:**

I read with interest this nicely written paper and I think that there is good evidence towards the authors' conjecture. Furthermore, the authors strive to produce some intuitive rationale to explain why heterogeneity leads to more structured language. I have a generally positive view of the work but, as a non-expert, I saw some weaknesses in the paper.

The most important objective of the paper is to demonstrate that a larger heterogeneous population leads to higher quality language. I was expecting a large variety of scenarios showing the impact of this heterogeneity. However, a single scenario is considered for the training speed, by sampling p_i from a log-normal(eta, sigma) with eta=-1 and sigma=1/2. These parameters do not lead to large heterogeneity. The distribution of p_i has expectation equal to exp(-1+(0.5^2)/2)=0.42 with variance equal to 0.05. This seems a very concentrated distribution. Given the main objective of this paper, it is interesting to see what is the impact of more dispersed distributions. Maybe not only varying the log-normal parameters to create a large variance but even considering other distributions such as the more natural Beta(a,b) distribution for the parameter p_i (which is a probability). What would be the impact on the increasing trends of neg-entropy and synchronization (see Figure 4) of having larger heterogeneity in the population?

Again, given that the main objective is to show that larger heterogeneous populations leads to higher quality language, I missed a larger variation on the population size. It increases linearly from N=2 to N=20 (meaning 10 senders and 10 receivers). It is not possible to simulate with much larger populations, with a geometrical increase? We are likely to see a more clear effect if populations change in orders of magnitude. I missed a larger range for his variation.

In page 8 and wrt Figure 4, the authors say that "compositionally and generalization have a small positive but statistically significant correlation with Spearman coefficients above 0.3 (p-values> 0:05)". This is not correct. Appendix C shows that, for generalization in heterogeneous populations, the Spearman correlation index is equal to -0.07, a negative and very low value. It is hard to believe that this low value is statistically significant and, worse yet, it is negative. Indeed, there is not a visually clear presence of a trend with the N increase in the compositionality and generalization plots in Figure 4.

Also, the statistical test is significant when the p-value is small, traditionally smaller than 0.05. In the excerpt copied above and in several other passages throughout the paper, the authors mention that statistical significance corresponds to having a p-value larger than 0.05. It should be smaller, not larger.



**Summary Of The Paper:**

This paper aims at solving a conflicting empirical observation and the present-state models for emerging languages. It has been observed that larger populations produce more structured languages. However, the state-of-art neural-based models have not been able to generate languages with such characteristics. This paper shows that a key ingredient is to allow population heterogeneity in the neural models rather than the current identically distributed specifications.

**Summary Of The Review:**

I have a generally positive view of the work but, as a non-expert, I saw some weaknesses in the paper.

---

> ### Author Response · Authors · 2021-11-18
> **Response to Reviewer 5n9H (1/1)**
>
> *We would like to thank Reviewer 5n9H for her/his useful comments. Reviewer 5n9H provides very constructive remarks with lots of technical insights. In particular, Reviewer 5n9h comment on our distribution has triggered an additional experiment that may be a clue to better understand the lack of significance of one of our results. We answer each bullet point providing more details/additional experiments/corrections:*
>
> > The most important objective of the paper is to demonstrate that a larger heterogeneous population leads to higher quality language. I was expecting a large variety of scenarios showing the impact of this heterogeneity. However, a single scenario is considered for the training speed, by sampling p_i from a log-normal(eta, sigma) with eta=-1 and sigma=1/2. These parameters do not lead to large heterogeneity. The distribution of p_i has expectation equal to exp(-1+(0.5^2)/2)=0.42 with variance equal to 0.05. This seems a very concentrated distribution. Given the main objective of this paper, it is interesting to see what is the impact of more dispersed distributions.
>
> We totally agree on this point. Actually, the heterogeneity plot is the hardest experiment to run because we have to build quite large populations, for several seeds and alter agent learning speeds (leading to longer trainings). That’s why we did not consider lots of scenarios.
>
> However, you raise a fair point to mention that the distribution considered in the paper may be too concentrated as illustrated. Following your remark and our previous observations, we launched the same experiment with a higher standard deviation ($\sigma_{p}=1$).
>
> Results are presented in the General Response and included as __new Figure 4__.
>
> > Maybe not only varying the log-normal parameters to create a large variance but even considering other distributions such as the more natural Beta(a,b) distribution for the parameter p_i (which is a probability). What would be the impact on the increasing trends of neg-entropy and synchronization (see Figure 4) of having larger heterogeneity in the population?
>
> The experiment with the Beta(1,2) is currently running **(UPDATE: the experiment has finished. We added results and comments)**. We will provide the result in the discussion as soon as possible. Following this remark and the previous one, we added __Appendix F.2__ in which we discuss the impact of the distribution with those two scenarios (standard deviation change ; distribution change).
>
> > Again, given that the main objective is to show that larger heterogeneous populations leads to higher quality language, I missed a larger variation on the population size. It increases linearly from N=2 to N=20 (meaning 10 senders and 10 receivers). It is not possible to simulate with much larger populations, with a geometrical increase? We are likely to see a more clear effect if populations change in orders of magnitude. I missed a larger range for his variation.
>
> First a small remark on how we defined N. Actually, we set in the paper: N=number of speakers = number of listeners. Therefore, when N=20, we have 20 senders and 20 receivers. That said, your remark still holds.
> Actually, it is difficult for us to run simulations with a geometric scale because experiments are actually long to run. It is mainly explained because:
> we assume a fully connected communication graph, meaning that when N=20, we have actually 400 pairs to update.
> We play with agent learning speeds by altering agents' learning steps. It therefore slows down the experiments.
>
> __Note__: In his review, Reviewer UKkL mentioned that it is the  “Largest-scale study of the impact of population size on emergent language structure” (at least before ICLR2022 submissions).
>
> > In page 8 and wrt Figure 4, the authors say that "compositionally and generalization have a small positive but statistically significant correlation with Spearman coefficients above 0.3 (p-values> 0:05)". This is not correct. Appendix C shows that, for generalization in heterogeneous populations, the Spearman correlation index is equal to -0.07, a negative and very low value. It is hard to believe that this low value is statistically significant and, worse yet, it is negative. Indeed, there is not a visually clear presence of a trend with the N increase in the compositionality and generalization plots in Figure 4.
>
> Thank you for noticing this mistake. It is actually an error when reporting the values from the Table. By including the new figure 4, we actually update this sentence as presented in the General Response.
>
> > Also, the statistical test is significant when the p-value is small, traditionally smaller than 0.05. In the excerpt copied above and in several other passages throughout the paper, the authors mention that statistical significance corresponds to having a p-value larger than 0.05. It should be smaller, not larger.
>
> It is a typographic error we have propagated throughout the manuscript. We updated it.

---

> > ### Comment · Reviewer_5n9H · 2021-11-20
> > **Reacting to the authors' comments**
> >
> > I appreciated your responses to my concerns. I had a positive view of this paper and I remain on the positive side. The minor points that I raised were properly addressed by the authors. My main concern was the scale of this study, which is not large. However, as reviewer UKkL mentioned and the authors emphasized, this is the “largest-scale study of the impact of population size on emergent language structure”.

---

### Official Review · Reviewer_UKkL · 2021-11-01

**Correctness:** 3
**Technical Novelty And Significance:** 3
**Empirical Novelty And Significance:** 3
**Recommendation:** 6
**Confidence:** 4

**Main Review:**


# Summary

This paper analyzes the structure of emergent languages in signaling games played by _populations_ of agents, motivated by a rich body of socio- and psycho-linguistics results showing that languages spoken by more people (and with more second-language learners) tend to be grammatically simpler (e.g. have a more impoverished morphology).  The authors measure various properties of the emergent languages as proxies for how "systematic" a language is, and show several things.  First, increasing the number of agents does not correlate with any of their measures, so population size alone does not suffice.  Second, in a minimally small population, various measures of "diversity" of the two agents do correlate with their systematicity measures.  Finally, in a large but diverse population, we do see correlations with population size and _some_ of the systematicity measures.  The paper is interesting and timely, and reports on a large number of experiments.  While I think the experiments could be more closely linked to the hypotheses from the sociolingusitic literature, the paper will be of interest to many researchers in emergent communication, NLP, and cognitive science, and could spur future work in this intersection.


* Strengths:
	- Thorough and competently executed experiments
	- Largest-scale study of the impact of population size on emergent language structure

* Weaknesses:
	- The experiments could be motivated more precisely from the cited literature (see comments below)
	- Experimental results were not always clearly reported (see comments below)



# Comments / questions:

* The cited Raviv et al studies crucially manipulated not just population size, but also _network structure_ (e.g. scale-free vs complete graphs). While it is completely fine for the present paper to only focus on population size and the homogeneity of the population, I would encourage the authors to at least mention this important extra dimension of the Raviv et al papers.  As presently written, the reader can get the impression that those studies only look at population size.  This could also be re-mentioned in Section 3.2, where the authors helpfully point out some of their modeling assumptions.

* I would also encourage the authors to cite Wray and Grace 2007 "The consequences of talking to strangers" (https://www.sciencedirect.com/science/article/pii/S0024384105000999) in their related work section.

* Why did the authors choose the full object-reconstruction loss as opposed to a choose-among-distractors loss (as is slightly more common on the emergent communication literature)?  Did they try the latter and find similar results, or do the results depend on the object-reconstruction setting?

* Was there a reason not to use the (straight-through) Gumbel Softmax trick for training here?

* Presentation of Entropy in \S3.3: (i) equation (5) is negative entropy, as the authors note at the end of the paragraph.  But the equation is introduced as just being entropy.  I would encourage them to make the definition (e.g. of $h$) include the negative sign, so that it's normal entropy, and then just mention that they report negative.  (ii) I don't know what it means to say that minimizing entropy "reinforces the information bottleneck principle", since the latter is about two forms of mutual information.  I'd like to either hear something more explicit/detailed here or have it dropped.

* Reporting results: (i) I would like to see actual numbers and a statistical analysis for e.g. the results in 5.1 / Fig 1, even if in an appendix (as is done in S5.3).  (ii) Similarly, they write: "In addition, we do not observe any language systematicity. By systematicity, we mean languages whose properties are stable across seeds."  It's hard to know what exactly the authors did and are reporting here, so a little more detail would be welcome.  (iii) I think a baseline of randomly-initialized / untrained agents for these metrics would be informative to see.

* p 7: "language neg-entropy is reaching almost 0, meaning almost a bijective language mapping".  I might be misunderstanding, but won't your entropy also be minimized by a _uniform_ language, that outputs the same sequence of symbols with probability 1 for every object?  Since the entropy is taken over next-tokens in the sequence in average over each object, I don't see any pressure for distinct messages for distinct objects in that measure, which is what a bijective mapping sounds like to my ear.  In other words: doesn't your measure of entropy just measure how _deterministic_ the mapping from objects to messages is, not necessarily how bijective it is?  I may be misunderstanding, but it would help to clarify if so.


# Typographic comments:

* p 3: "community size and leaning properties" --> "community size and learning properties"

* p 4, eqn (4): "$m_i \sim \pi_{\theta_1}(\cdot | v)": the "\theta_1" here should be "\theta_2"

* Caption on Fig 2: the "x" in "x-axis" should be in math mode

* page 7: the quotation marks around "fast enough" should be fixed to ``fast enough'' (and same for "ease-of-teaching")

* page 9: "actually be to simplistic" --> "actually be too simplistic"

**Summary Of The Paper:**

This paper analyzes the structure of emergent languages in signaling games played by _populations_ of agents, motivated by a rich body of socio- and psycho-linguistics results showing that languages spoken by more people (and with more second-language learners) tend to be grammatically simpler (e.g. have a more impoverished morphology).  The authors measure various properties of the emergent languages as proxies for how "systematic" a language is, and show several things.  First, increasing the number of agents does not correlate with any of their measures, so population size alone does not suffice.  Second, in a minimally small population, various measures of "diversity" of the two agents do correlate with their systematicity measures.  Finally, in a large but diverse population, we do see correlations with population size and _some_ of the systematicity measures.  The paper is interesting and timely, and reports on a large number of experiments.  While I think the experiments could be more closely linked to the hypotheses from the sociolingusitic literature, the paper will be of interest to many researchers in emergent communication, NLP, and cognitive science, and could spur future work in this intersection.

**Summary Of The Review:**

The paper provides a rich set of experiments looking at the effect of population size and diversity of emergent languages, and so will be of interest to many working at the intersection of emergent communication in NLP and cognitive science.

---

> ### Author Response · Authors · 2021-11-18
> **Response to Reviewer UKkL (1/2)**
>
> *We would like to thank Reviewer UKkL for her/his useful comments. We provide more details and explanations for each comment:*
>
> > The cited Raviv et al studies crucially manipulated not just population size, but also network structure (e.g. scale-free vs complete graphs). While it is completely fine for the present paper to only focus on population size and the homogeneity of the population, I would encourage the authors to at least mention this important extra dimension of the Raviv et al papers. As presently written, the reader can get the impression that those studies only look at population size. This could also be re-mentioned in Section 3.2, where the authors helpfully point out some of their modeling assumptions.
>
> We are aware that Raviv et al. papers extend the research on populations way beyond population size. To better highlight the variety of Raviv et al. works, we updated the passage mentioning Raviv et al. insisting on the variety of their works as shown in the General Response __(see updates in the new Related Section and changes in Appendix G.1.2.)__.
>
> > I would also encourage the authors to cite Wray and Grace 2007 "The consequences of talking to strangers" (https://www.sciencedirect.com/science/article/pii/S0024384105000999) in their related work section.
>
> The reference you mentioned is indeed very relevant. As discussed in the General Response, we added it in the conclusion when we mention that other social factors could influence language properties. This update is shown in __Appendix G1.3__.
>
> > Why did the authors choose the full object-reconstruction loss as opposed to a choose-among-distractors loss (as is slightly more common on the emergent communication literature)? Did they try the latter and find similar results, or do the results depend on the object-reconstruction setting?
>
> When building our setting, we followed the protocol described in Chaabouni et al. (2020) [1] because they provide an extensive analysis of compositionality and generalization allowing us to compare the score we get with their values. As mentionned in the paper, using a discrimination loss adds new setup parameters (such as the number of discriminator) that can have a huge impact on the scores. Indeed, the problem becomes more complicated when the number of discriminators becomes large enough (compared to the input space size) ; while it is not even necessary to introduce compositionality patterns when the number of discriminators is very small compared to the input space size.
>
> As mentioned on page 3, this design choice does not undermine our conclusions. To fully support this claim, we are running a subset of our experiments with the InfoNCE loss, but they may not be over before the end of the rebutal.
>
> [1] Chaabouni, R., Kharitonov, E., Bouchacourt, D., Dupoux, E., & Baroni, M. (2020). Compositionality and Generalization In Emergent Languages. In Proceedings of the 58th Annual Meeting of the Association for Computational Linguistics (pp. 4427-4442).
>
> > Was there a reason not to use the (straight-through) Gumbel Softmax trick for training here?
>
> Gumbel-Softmax tricky provides a robust way to perform this kind of optimization (Jang et al. 2017 [1]) and has been used in lots of emergent communication works (e.g. Havrylov & Titov 2017 [2], Mordatch & Abbeel 2018 [3]). Here, we did not face any major optimization issue when using RL, thus GSoftmax was not required. More importantly, we think that RL better fits the human learning process in communication where there is no direct backpropagation between a speaker and a listener. As we are linking our work with socio-linguistics, it sounds like a more reasonable optimization choice.
>
> [1] Jang, E., Gu, S., & Poole, B. (2017). Categorical reparameterization with Gumbel-Softmax. In Proceedings of ICLR Conference Track, Toulon, France.
>
> [2] Havrylov, S., & Titov, I. (2017). Emergence of language with multi-agent games: Learning to communicate with sequences of symbols. arXiv preprint arXiv:1705.11192.
>
> [3] Mordatch, I., & Abbeel, P. (2018). Emergence of grounded compositional language in multi-agent populations. In Thirty-second AAAI conference on artificial intelligence.
>
> > Presentation of Entropy in \S3.3: (i) equation (5) is negative entropy, as the authors note at the end of the paragraph. But the equation is introduced as just being entropy. I would encourage them to make the definition (e.g. of h) include the negative sign, so that it's normal entropy, and then just mention that they report negative. (ii) I don't know what it means to say that minimizing entropy "reinforces the information bottleneck principle", since the latter is about two forms of mutual information. I'd like to either hear something more explicit/detailed here or have it dropped.
>
> (i) We inserted your suggestion as we agree it will improve clarity.
>
> (ii) We dropped the sentence about the Information Bottleneck Principle as it does not help comprehension.

---

> > ### Author Response · Authors · 2021-11-18
> > **Response to Reviewer UKkL (2/2)**
> >
> > > Reporting results: (i) I would like to see actual numbers and a statistical analysis for e.g. the results in 5.1 / Fig 1, even if in an appendix (as is done in S5.3). (ii) Similarly, they write: "In addition, we do not observe any language systematicity. By systematicity, we mean languages whose properties are stable across seeds." It's hard to know what exactly the authors did and are reporting here, so a little more detail would be welcome. (iii) I think a baseline of randomly-initialized / untrained agents for these metrics would be informative to see.
> >
> > (i) For Figure 1 of Section 5.1, the statistical analysis was reported in Appendix C - Table 2 (Homogeneous Population) but we did not mention those values in the main text. For a sake a clarity, we updated the first paragraph of Section 5.1 where the Figure 1 is described, including the statistical analysis __(difference is displayed in Appendix G3.1)__.
> >
> > (ii) By this sentence, we equate “systematicity” to the variation of the scores across seeds. We thus measure “systematicity” via the standard deviation of the metrics across seeds. In the context of language the term systematic may indeed be confusing, we have therefore replaced all the word “systematic” by “stable” and “systematicity” by “stability” and replaced the sentence “By systematicity, we mean languages whose properties are stable across seeds.” by “By stable languages, we mean languages whose properties have low variations across seeds for a given population size.”.
> >
> > (iii) Random baseline actually tends to rescale our plots and make the plots harder to read. However, Reviewer UKkL is true to mention that random baselines could be informative. Therefore, we added a table in the __new Appendix A__ indicating the values of the four metrics with randomly initialized/untrained agents.
> >
> > > p 7: "language neg-entropy is reaching almost 0, meaning almost a bijective language mapping". I might be misunderstanding, but won't your entropy also be minimized by a uniform language that outputs the same sequence of symbols with probability 1 for every object? Since the entropy is taken over next-tokens in the sequence in average over each object, I don't see any pressure for distinct messages for distinct objects in that measure, which is what a bijective mapping sounds like to my ear. In other words: doesn't your measure of entropy just measure how deterministic the mapping from objects to messages is, not necessarily how bijective it is? I may be misunderstanding, but it would help to clarify if so.
> >
> > There is a lack of clarity in this sentence. You are right to note that entropy measures how deterministic the mapping from objects to messages is. Therefore, if the entropy is equal to 0, the speaker sends a unique message with probability 1 for every object. Then, two cases are possible:
> >
> > - There is a distinct message for each distinct object: bijective mapping.
> >
> > - There could exist messages referring to the same object (even if we have entropy 0) as you mentioned.
> >
> > However, our experiments all reach 100% training accuracy. It implies that the second case is impossible, otherwise the listener could not succeed every time. That said, we agree that this sentence may be ambiguous and that it does not add crucial content to the paper.
> > We have thus removed the part of the sentence “meaning almost a bijective language mapping”.
> >
> > > Typographic comments
> >
> > We have corrected all the typographic errors mentioned by Reviewer UKkL. We thank Reviewer UKkL for taking time to mention them.

---

> > > ### Comment · Reviewer_UKkL · 2021-11-22
> > > **Many thanks for the detailed reply and revisions**
> > >
> > > I was already a fan of this paper, but had some minor (mostly presentational) concerns, all of which have been very adequately addressed in this discussion and the revised version, so my view of the paper remains very positive.  Thank you to the authors for all their hard work!

---

### Official Review · Reviewer_zxeP · 2021-11-02

**Correctness:** 2
**Technical Novelty And Significance:** 4
**Empirical Novelty And Significance:** 3
**Recommendation:** 6
**Confidence:** 4

**Main Review:**

Strengths:
- This is a well-written paper with a very engaging flow.
- This paper provides important insights for language emergence research in machine learning.

Weaknesses:
- The largest effects of heterogeneity are on speakers synchronization and negative entropy; there is low, no, or negative effect on compositionality and generalization. The paper is overclaiming to say that the emergent language is more structured with heterogeneity when the compositionality and generalization (metrics we arguably care about more) are not significantly affected with increasing population or speaker-listener asymmetry (when $\rho_\bullet \geq 10^0$).
- The claim that network capacity is a confounding factor of training speed relies on the speed of achieving 95% training accuracy (related: Figure 2 left of [1]). However, is this sufficient to claim that the effect is equivalent in the setting you are studying? How models fit training data and how they generalize to unseen data can have different dynamics. Do these "equivalent" models (fast model with high model capacity and slow model with low model capacity) achieve similar values on the four language property metrics in the paper when they get to 95% training accuracy?

Other questions:
- When you split the data into train and test, did you try splitting them systematically? It would be interesting to evaluate whether the emergent language can generalize OOD.
- I find this hypothesis interesting: "extreme agents, especially slow listeners, behave as kinetic bottlenecks, forcing the speakers to structure their languages. Thus, language emergent properties would be particularly determined by fast speakers and/or slow listeners." Is there any intuition for why the relative speed differences would have this effect?

Suggestions:
- In Section 3.4 Control parameters of local asymmetry: Typo: should be $p^S$ instead of $p_S$ in the fifth line of the paragraph.
- In Section 3.4 Distributing Population Heterogeneity: "update probability factor" in the last line is not clear until later in the paper.
- Figure 3 caption: after "pretrained with a speaker", add "(resp. listener)"?
- Section 5.2: Typo: "statically" -> "statistically".

References:
[1] Kaplan, J., McCandlish, S., Henighan, T., Brown, T.B., Chess, B., Child, R., Gray, S., Radford, A., Wu, J. and Amodei, D., 2020. Scaling laws for neural language models. arXiv preprint arXiv:2001.08361.

**Summary Of The Paper:**

The authors refer to prior work in sociolinguistic literature to state that larger communities create more systematic languages. However, they point out, this apparent correlation between language structure and population size has evaded machine learning practitioners studying language emergence. This paper claims that populations explored in machine learning have largely been homogeneous and that population heterogeneity is key to the emergence of structure in artificial agents. The authors reproduce the failure to achieve systematic languages by scaling up the population and then show that they can indeed get more structure by introducing heterogeneity. They explore introducing asymmetry between the speaker and the listener based on model capacity and learning speed, leading to an increase in language structure when the speaker is faster or has more capacity. The authors note that this effect only depends on the relative differences between the speaker and the listener and not the absolute values (there is a correlation with absolute values but the variation has a low magnitude). Further, they show that larger networks need fewer epochs to reach similar training accuracy, concluding that network capacity is a confounding factor of training speed in their setting. Finally, the authors create a population of heterogeneous agents by imbibing them with different learning speeds by updating an agent $i$ with probability $p_i$ after each round of the Lewis game. Four properties of the emergent language are studied: speakers synchronization, (negative) conditional entropy given an object, topographic similarity, and generalization. These metrics either improve or remain approximately at the same values as the population size increases.

**Summary Of The Review:**

I believe this is an important paper to encourage future research into heterogeneity as a component for language emergence. The paper is well-written and the authors are able to scale up the Lewis game setting without observing a decline in desirable language properties. However, the paper in its current form makes claims that cannot be substantiated in the evidence, which I find to be a significant barrier for me to recommend acceptance.

UPDATE AFTER REBUTTAL:

After the discussion with the authors, the paper has toned down its claims and added clarity about what it does not claim: it does not present an overall better language emergence setup, and the goal is to primarily address the apparent decline in structure as population size goes up in machine learning in contrast to sociolinguistic studies. Thus, I am increasing my score for this paper.

---

> ### Author Response · Authors · 2021-11-18
> **Response to Reviewer zxeP (1/2)**
>
> *We would like to thank Reviewer zxeP for all her/his positive compliments and the relevance of her/his remarks. All the questions imply additional experiments or explanations.  We provide more details, additional experiments and discussion for each remark/question:*
>
> > The largest effects of heterogeneity are on speakers synchronization and negative entropy; there is low, no, or negative effect on compositionality and generalization. The paper is overclaiming to say that the emergent language is more structured with heterogeneity when the compositionality and generalization (metrics we arguably care about more) are not significantly affected with increasing population or speaker-listener asymmetry (when ρ∙≥1).
>
> Behind this remark, we see two sub-points that can be discussed:
>
> - __about the significance of the results for compositionality and generalization when increasing speaker-listener asymmetry__: Reviewer zxeP notices that there is no compositionality/generalization improvement when ρ∙≥1. Starting from ρ∙=1, the metrics effectively stagnate. However, we do not see this stagnation as a negative or insignificant effect. Indeed, metrics have already reached a high bound: speaker synchronization is close to 0.8/0.9 ; neg-entropy almost equal to 0, compositionality is almost equal to 0.7 in mean and test accuracy almost optimal. On the contrary, when ρ∙<<1, those metrics reach low values. Therefore, we do not think that this stagnation is negative. However, we agree that we can wonder why the stagnation seems to start for ρ∙=1. Actually, it is due to the initial agents parameters we chose (architectures, learning rates, ...). For example, curves are translated when we variate agent hidden sizes (speaker hidden size=64 and listener hidden size=512). You can see the plot in new Appendix C. In this plot, the point of stagnation is not ρ∙=1 anymore. Actually, there are a lot of initial factors that indirectly affect agents' learning speeds (architectures, optimization, etc.). Therefore, it is likely that this stagnation point varies depending on all the parameter choices.
> Following your remarks, we added __Appendix C__ and added a __comment in Section 5.2 (the difference is shown in Appendix H2.1)__.
>
>
> - __about the significance of the results for compositionality and generalization when increasing population__: See General Response.
>
> > The claim that network capacity is a confounding factor of training speed relies on the speed of achieving 95% training accuracy (related: Figure 2 left of [1]). However, is this sufficient to claim that the effect is equivalent in the setting you are studying? How models fit training data and how they generalize to unseen data can have different dynamics. Do these "equivalent" models (fast model with high model capacity and slow model with low model capacity) achieve similar values on the four language property metrics in the paper when they get to 95% training accuracy?
>
> We think that our sentence “We conclude that network capacity is a confounding factor of training speed in our setting.” in the main text may be too strong. Indeed, we did not intend to say that capacity / learning step ratios are equivalent. Indeed, increasing capacity may lead to other phenomena not captured by the learning speed. Our goal was to give a hypothesis that could explain (i) why we observe this trend with capacity ; (ii) why the trends are similar. With the plot in Figure 3, there is first evidence that allows us to think that capacity may indirectly affect learning speed and then explaining what we see. However, we do not think that there is any kind of equivalence and part of the effects observed with the capacity could not be explained only with an indirect influence on learning speed. Following your remark, we updated the sentence “We conclude that network capacity is a confounding factor of training speed in our setting.” in __Section 5.2 (the difference is shown in Appendix H2.2)__ to better insist that we do not intend to say that the models are “equivalent”.

---

> > ### Author Response · Authors · 2021-11-18
> > **Response to Reviewer zxeP (2/2)**
> >
> > > When you split the data into train and test, did you try splitting them systematically? It would be interesting to evaluate whether the emergent language can generalize OOD.)
> >
> > To measure generalization we followed the protocol described in Kottur et al. (2017) [1] and Chaabouni et al. (2020) [2]. If we understand well, the experiment you propose is analogous to Cogswell et al. (2019)’s measure of generalization [3].
> >
> > We ran the experiments with this construction of the test set. However, nothing better/worse appended with this split. Indeed, the trends were the same with a top test accuracy of 75% (when working with 4 attributes).
> >
> > [1] Satwik Kottur, Jose ́ Moura, Stefan Lee, and Dhruv Batra. Natural language does not emerge ‘naturally’ in multi-agent dialog. In Proc. of Empirical Methods in Natural Language Processing (EMNLP), 2017.
> >
> > [2] Rahma Chaabouni, Eugene Kharitonov, Diane Bouchacourt, Emmanuel Dupoux, and Marco Baroni. Composi- tionality and generalization in emergent languages. In Proc. of the Association for Computational Linguistics (ACL), 2020.
> >
> > [3] Michael Cogswell, Jiasen Lu, Stefan Lee, Devi Parikh, and Dhruv Batra. Emergence of compositional language with deep generational transmission. arXiv preprint arXiv:1904.09067, 2019.
> >
> >
> > > I find this hypothesis interesting: "extreme agents, especially slow listeners, behave as kinetic bottlenecks, forcing the speakers to structure their languages. Thus, language emergent properties would be particularly determined by fast speakers and/or slow listeners." Is there any intuition for why the relative speed differences would have this effect?
> >
> >
> > We showed in Section 5.2 that our metrics are highly influenced by the relative learning speed of speakers/listeners. Especially, when ρ is large enough (right part of the curves on Figure 2) (eq. fast speakers/low listeners) speakers start to synchronize and develop more structured languages (lower entropy, higher compositionality and generalization).
> > The success of an experiment is dependent on both the adaptation of the speaker and the listener. Qualitatively, when most of the co-adaptation is performed by the listener (the speaker is a kind of “kinetik bottleneck”), success can be achieved even with an almost degenerated language (the only condition is that the speaker develops an unambiguous language). On the contrary, when the listener is the “kinetik bottleneck”, the speaker develops a language that better fits the online comprehension of the listener at any time. If the listener is slow enough, speakers have time enough to find the messages that best fit the listener's comprehension (messages providing highest rewards) and therefore synchronize on the same messages and develop low entropy languages. Additionally, we interpret the gain of compositionality to the fact that compositional languages are easier to learn for neural agents [4]. Since most of the co-adaptation work is performed by the Speaker, there may be an indirect pressure to produce languages easy to understand. Eventually, the gain of generalization can be interpreted as a side effect of the gain of language structure.
> >
> > Once heterogeneities are distributed into the populations (Section 5.3 - Figure 4), each speaker has to build a language that fits both the fast and slow listeners. Fast listeners are not restrictive as they can perform the co-adaptation quickly while slow listeners impose larger co-adaptation from the speakers. Therefore, we make the assumption (causality is not demonstrated in the paper) that the slow listeners into the population are the ones imposing low entropy, speaker synchronization and compositionality to the speakers.
> >
> > [4] Fushan Li and Michael Bowling. Ease-of-teaching and language structure from emergent communication. In Proc. of Advances in Neural Information Processing Systems (NeurIPS), 2019.
> >
> >
> > > - In Section 3.4 Control parameters of local asymmetry: Typo: should be pS instead of pS in the fifth line of the paragraph.
> > > - In Section 3.4 Distributing Population Heterogeneity: "update probability factor" in the last line is not clear until later in the paper.
> > > - Figure 3 caption: after "pretrained with a speaker", add "(resp. listener)"?
> > > - Section 5.2: Typo: "statically" -> "statistically".
> >
> > We corrected all the typographic errors mentioned by the reviewer. We thank the reviewer for taking time to mention them.

---

> > > ### Comment · Reviewer_zxeP · 2021-11-19
> > > **Response to the authors**
> > >
> > > Thank you for taking the time to respond to my concerns.
> > >
> > > > If we have not fully demonstrated how population diversity can reproduce the non neural results, at least we have demonstrated that it is no longer warranted to do population studies with identical neural agents.
> > >
> > > I do think the message about showing benefits over populations of homogenous agents is an important one. But for this, the paper makes it very difficult to compare the plots for the scores of homogenous and heterogeneous populations by being spread out across four pages (Figure 1 and Figure 4). Since this is a crucial message of the paper, I would strongly suggest including a single figure that plots both the homogenous and the heterogeneous scores. This would enable a more accessible discussion of the differences between the homogenous and heterogeneous results.
> > >
> > > Finally, I also think it is important for the paper to clarify that in comparison to the general setting explored in the language emergence literature of N=2 with symmetric agents, the resulting compositionality from a population of heterogeneous agents is comparable and the generalization is worse. Thus, you are not currently claiming an overall better language emergence setup than those commonly studied in the literature. But instead, the message is primarily to address a long-standing problem in scaling up populations.
> > >
> > > I would be inclined to increase my score if there is a clearer portrayal of what the paper's results are actually reporting.

---

> > > > ### Author Response · Authors · 2021-11-21
> > > > **Follow-up on Reviewer zxeP's response**
> > > >
> > > > > I do think the message about showing benefits over populations of homogenous agents is an important one. But for this, the paper makes it very difficult to compare the plots for the scores of homogenous and heterogeneous populations by being spread out across four pages (Figure 1 and Figure 4). Since this is a crucial message of the paper, I would strongly suggest including a single figure that plots both the homogenous and the heterogeneous scores. This would enable a more accessible discussion of the differences between the homogenous and heterogeneous results.
> > > >
> > > > First of all, we thank the reviewer for his *excellent idea*. We **updated figure 4** to overlap homogenous, heterogenous and best agents language properties. In short, the figure now better illustrates that language metrics are correlated to population size in heterogeneous communities, and it does not insidiously claim that heterogeneous populations lead to a higher quality emergence language overall.
> > > >
> > > > In more details: on Figure 4, we see that language properties of homogeneous populations with $\rho_{speed}=1$ decrease when population size is increasing. On the contrary, metrics get higher for heterogeneous populations (generalization stays almost constant). Thus, when modeling large populations, there is a gain distributing heterogeneities within populations.
> > > > However, metrics remain below the scores of the best pair observed for $N=2$ when varying $\rho_{speed}$. It means that enlarging populations does not lead to an absolute better language than the best one we can obtain with a minimal size population. Though, when population size increases, heterogeneous populations metrics get closer to the best pair scores. Therefore, population size tends to synchronize agents on a language whose properties are close to language of a single pair.
> > > >
> > > > > Finally, I also think it is important for the paper to clarify that in comparison to the general setting explored in the language emergence literature of N=2 with symmetric agents, the resulting compositionality from a population of heterogeneous agents is comparable and the generalization is worse. Thus, you are not currently claiming an overall better language emergence setup than those commonly studied in the literature. But instead, the message is primarily to address a long-standing problem in scaling up populations.
> > > >
> > > > Following your remark, **we added a paragraph at the end of Section 5.3 named "On the overall performances of heterogeneous population"**. We present what it is just said above.
> > > >
> > > > We think it will now make it clearer that the aim of the paper  is not to claim a better language emergence setup but rather to discuss a well-documented correlation observed in sociolinguistics.

---

> > > > > ### Comment · Reviewer_zxeP · 2021-11-23
> > > > > **Updating my score**
> > > > >
> > > > > Thank you for your updates, the updated figures and text add a lot of clarity. It would be nice to discuss the limitations of the heterogeneous population game as a general language emergence setup in the conclusion as a future direction to explore, but I leave this decision up to the authors.
> > > > >
> > > > > I am updating my score because my major concerns were addressed; however, the results are still quite modest with respect to the claims that "larger heterogeneous populations lead to increased structure."

---

> > > > > > ### Author Response · Authors · 2021-11-23
> > > > > > **Update in the conclusion**
> > > > > >
> > > > > > Thanks for all the useful comments during the discussion. All of them were very helpful to improve the paper.
> > > > > >
> > > > > > As you suggested, we added a sentence insisting on the limits of heterogeneous populations game as a general improvement of emergent languages with the space left in the conclusion:
> > > > > >
> > > > > > *"Fourth, although heterogeneous populations recover a socio-linguistics result, the average scores remain below the best emergent protocol. We leave for future work how heterogeneous populations may be leveraged to further structure the language (e.g. more complex tasks, larger population)."*
> > > > > >
> > > > > > This sentence is part of our list of potential follow-ups and improvements of our model. It thus supplements our list of future directions linked to the paper.

---

### Author Response · Authors · 2021-11-18
**General Response for all the reviewers (1/2)**

We first want to thank the reviewers for the high quality of their responses that are both relevant and well documented. We appreciated the diversity of comments ranging from high level non-ML remarks to very specific details on statistical significance. We would like to start the discussion even if some complementary experiments are still running. We will subsequently add the remaining experimental results as soon as possible. Changes in the manuscript have been colored in red and the differences between the old version and the new one are all reported in **Appendix G** of the new manuscript.

We hope to address the *two main issues* of the reviewers in this General Response about the *literature* review and the *significance of the results*:

*I - We explain the position of the paper with respect to the literature and propose updates that include references suggested by the reviewers. __See Related Section and conclusion in the new manuscript.__*

*II - We explain why we think our results in the original form already contribute to the literature and we enforce Figure 4 with an additional experiment suggested by Reviewer 5n9H. Following Reviewer zxeP remark, we also added a paragraph enforcing that we do not claim a better language emergence setup with heterogeneous populations. It makes it easier for the reader to understand that we rather discuss a well-documented socio-linguistics correlation. __See new Figure 4 in the new manuscript and Section 5.3 paragraph "On the overall performances of heterogeneous populations".__*

Although we expect those new elements to strengthen our work, they do not change the claim, nor the storyline of the paper, and thus remain within the scope of minor paper revisions.*

## I - Literature review

First, we want to clarify the position of our paper relative to past works.

The main point of  the paper is to show that population modeling with neural agents is typically done with too simplistic assumptions. We demonstrate that a well-known socio-linguistics correlation cannot be recovered with current agent modeling (Section 5.1 - Figure 1). In this context, we study heterogeneities and we show that controlling learning speed asymmetry within a pair of agents tremendously affects language properties (Section 5.2 - Figure 2). Then, when distributing learning speed heterogeneities within the population, we recover some population effects (Section 5.3 - Figure 4). We thus advocate for the distribution of agent parameters when building neural populations for a more plausible modeling. Our paper is thus principally relevant to the neural language emergence field and especially to the group of papers that  scale experiments to larger populations of agents, and this is why we focus on this set of work in our Related Section. We hope, however, that the current results will incite neural modelers to pay more attention to works done in the non neural field of language emergence in order to validate their models against empirical phenomena as well as get inspiration regarding the basic assumptions of the models.

We totally agree that the (non-neural) works mentioned by Reviewer grQ9 have been fundamental for the study of language emergence as an empirical phenomenon and that our position with respect to this literature was not explicit enough. In order to better relate our paper to this non-neural literature, we have updated the __paragraph “Populations in simulated emergent communication”  of the Related Section (see the difference in Appendix G1.1).__

Additionally, following Reviewer UKkL remark, we have also updated a sentence in the __paragraph “Population size in sociolinguistics” of the Related Section to better insist on the diversity of Raviv et al. works. (see the difference in Appendix G1.2)__ .

Eventually, we completed the __conclusion__ including new (non-neural) references discussing the role of other social/population factors such as those mentioned by reviewers UKkL and grQ9. __(see the difference in Appendix G1.3)__

---

> ### Author Response · Authors · 2021-11-18
> **General Response for all the reviewers (2/2)**
>
> ## II - Significance of the results
>
> One shared comment by Reviewers zxeP, 5n9H and grQ9 is that the results in Figure 4 for compositionality and generalization may be weak. Thus, they mention that the paper is overclaiming when analyzing this result. We provide here a global response to this concern.
>
> We agree with the reviewers that the original Figure 4 did not display significant increases for compositionality and generalization: compositionality increase is low (slightly more significant in Figure 13 of Appendix F.2 (Figure 12-Appendix E in previous manuscript) for a different set of parameters) ; generalization is almost constant. However, we did not observe a drop in results as with the homogeneity population. Therefore, Figure 4 was showing a significant gain of speaker synchronization, neg-entropy and a stabilization of compositionality and generalization when increasing population size. If we have not fully demonstrated how population diversity can reproduce the non neural results, at least we have demonstrated that it is no longer warranted to do population studies with identical neural agents.
>
> Furthermore, following Reviewer Rn9H who notes that our actual distribution of learning steps is not very wide, we have re-run the experiments with a larger standard deviation of our log-normal distribution of learning steps. __Results are displayed in the new Figure 4 of the manuscript and the difference with the old version in Appendix G.1.4__. As can be seen, the effects are increased overall with still positive effects on speakers synchronization and neg-entropy (with faster increase) and positive effects on compositionality. Therefore, one hypothesis on the absence of increase for compositionality in the original Figure 4 is that our distribution was too concentrated. No gain of generalization is observed here. However, we do not see a decrease of generalization as was observed in the homogeneous population.
>
> Analyzing this additional experiment and following reviewers remarks, we have performed the following change:
>
> - __Update Figure 4__ with this new distribution and __update its analysis in Section 5.3 paragraph ”Larger heterogeneous population leads to higher quality language”__. Move the old Figure 4 into Appendix F.2 and comment how distribution changes affect the trends __(see difference in Appendix H1.4)__.
>
> - __Added baselines on Figure 4 (homogeneous populations + best pair when N=2)__ to better read the results and comment the overall performances of heterogeneous populations.
>
> - __Added the paragraph "On the overall performances of heterogeneous populations" in Section 5.3__ to describe heterogeneous populations performances. We hope it will make it clearer that we do not claim a better language emergence setup with heterogeneous populations. The purpose of the paper is rather to discuss a well-known socio-linguistics correlation.
>
> - In the __conclusion__: de-emphasize our claims and insist that we only focus on a specific aspect of population modeling, enforcing the idea that homogeneous populations are definitely not good models of language emergence, and that  heterogeneities need to be further explored. __(see difference in Appendix G1.5)__.

---

> > ### Author Response · Authors · 2021-11-21
> > **General Response for all the reviewers : UPDATE**
> >
> > **UPDATE after discussion with Reviewers**
> >
> > Following the discussion with Reviewer zxeP, we **updated Figure 4** including scores of homogeneous populations and the best pair observed with N=2. We also added a paragraph at the end of Section 5.3, comparing the overall performances of heterogeneous populations/homogeneous populations and best pair.  We insist that heterogeneous populations have better language scores than homogeneous populations for large population size but that they remain below the “best languages” we can get with a minimal size population. We only note that populations get closer to the best pair when pop.size is increasing.
> >
> > We think that this paragraph **enforces that we do not claim a better language emergence setup** with heterogeneous populations and make it easier for the reader to understand that we rather **discuss a well-documented socio-linguistics correlation**.
> >
> > Finally, as the paper starts exceeding the maximum number of pages. **We moved the section "Network capacity may be co-founded with training speed" in the appendix** only keeping the main ideas of this paragraph at the end of paragraph "Understanding learning speed asymmetries."
> >
> > We expect that this final version of the paper addresses the reviewers' remarks. **This rebuttal has been a real pleasure, and we very like the new resulting draft.**

---

### Decision · Program_Chairs · 2022-01-20

**Decision:**

Accept (Poster)

**Comment:**

The authors investigate the claim that agents in emergent communication games will converge to a symmetric homogenous state.  In particular, the authors show/argue for diversity in the population to close the gap between observed trends in neural agents and those expected when studying natural languages (e.g. around structure).  Reviewers were generally positive, though requested a number of rhetorical changes needed and additional literature.  These have been addressed.